# Cbln1 regulates axon growth and guidance in multiple neural regions

**Peng Han**[1☯], **Yuanchu She**[1☯], **Zhuoxuan Yang**[1], **Mengru Zhuang**[1], **Qingjun Wang**[1], **Xiaopeng Luo**[1], **Chaoqun Yin**[1], **Junda Zhu**[1], **Samie R. Jaffrey**[2]*, **Sheng-Jian Ji**[1]*

1 School of Life Sciences, Department of Neuroscience and Department of Biology, Brain Research Center, Shenzhen Key Laboratory of Gene Regulation and Systems Biology, Southern University of Science and Technology, Shenzhen, Guangdong, China, 2 Department of Pharmacology, Weill Cornell Medicine, Cornell University, New York, New York, United States of America

☯ These authors contributed equally to this work.
* srj2003@med.cornell.edu (SRJ); jisj@SUSTech.edu.cn (SJJ)

**Data Availability Statement:** The microarray data has been deposited to the Gene Expression Omnibus (GEO) with accession number GSE169448. All other relevant data are within the paper and its Supporting Information files.

## Abstract

The accurate construction of neural circuits requires the precise control of axon growth and guidance, which is regulated by multiple growth and guidance cues during early nervous system development. It is generally thought that the growth and guidance cues that control the major steps of axon development have been defined. Here, we describe cerebellin-1 (Cbln1) as a novel cue that controls diverse aspects of axon growth and guidance throughout the central nervous system (CNS) by experiments using mouse and chick embryos. Cbln1 has previously been shown to function in late neural development to influence synapse organization. Here, we find that Cbln1 has an essential role in early neural development. Cbln1 is expressed on the axons and growth cones of developing commissural neurons and functions in an autocrine manner to promote axon growth. Cbln1 is also expressed in intermediate target tissues and functions as an attractive guidance cue. We find that these functions of Cbln1 are mediated by neurexin-2 (Nrxn2), which functions as the Cbln1 receptor for axon growth and guidance. In addition to the developing spinal cord, we further show that Cbln1 functions in diverse parts of the CNS with major roles in cerebellar parallel fiber growth and retinal ganglion cell axon guidance. Despite the prevailing role of Cbln1 as a synaptic organizer, our study discovers a new and unexpected function for Cbln1 as a general axon growth and guidance cue throughout the nervous system.

## Introduction

The precise control of axon pathfinding is critical for the correct neural wiring during nervous system development. The stimulation of axon growth and regulation of axon guidance have been shown to require adhesion molecules, diffusible signals, and morphogens such as Netrins [1–7], Slits [8–10], Ephrins [11], Semaphorins [10,12], Draxin [13], Shh [14], Wnts [15], and BMPs [16,17]. These axon guidance molecules bind to their receptors in the axon growth cones to activate various signaling pathways that eventually change the cytoskeleton [18]. The

**Funding:** This work was supported by National Natural Science Foundation of China (https://www.nsfc.gov.cn/) (31871038 and 32170955 to S.-J.J.), Shenzhen-Hong Kong Institute of Brain Science-Shenzhen Fundamental Research Institutions (http://bcbdi.siat.ac.cn/) (2022SHIBS0002), High-Level University Construction Fund for Department of Biology (https://bio.sustech.edu.cn/) (internal grant no. G02226301), Science and Technology Innovation Commission of Shenzhen Municipal Government (http://stic.sz.gov.cn/) (ZDSYS20200811144002008), and NIH (https://www.nih.gov/) (R35NS111631 to S.R.J.). The funders had no role in study design, data collection and analysis, decision to publish, or preparation of the manuscript.

**Competing interests:** The authors have declared that no competing interests exist.

**Abbreviations:** A, anterior; A➔P, anterior to posterior; A.U., arbitrary unit; BafA1, Bafilomycin A1; CA, commissural axon; Cbln1, cerebellin 1; cKO, conditional knockout; CNS, central nervous system; Cre, Cre recombinase; CTB555, CTB-Alexa Fluor 555; Ctrl, control; D, dorsal; DCN, dorsal commissural neuron; DEG, differentially expressed genes; DIG, digoxigenin; dI1, dI2, dI3, dI4, dorsal interneuron 1, 2, 3, 4; FACS, fluorescence-activated cell sorting; FP, floor plate; GC, granule cell; GluD1, GluD2, glutamate receptor delta 1, 2; GPN, Glycyl-L-phenylalanine 2-naphthylamide; h, hours; IF, immunofluorescence; IGL, inner granule layer; LGN, lateral geniculate nucleus; ML, molecular layer; NFM, neurofilament; Nrxn, neurexin; OC, optic chiasm; OE, overexpression; P, posterior; PC, Purkinje cell; PFs, parallel fibers; RGC, retinal ganglion cell; rhCbln1, recombinant human Cbln1 protein; SEM, standard error of the mean; shCbln1, shRNA against Cbln1; shCtrl, control shRNA; shNrxn, shRNA against neurexin; St, stage (Hamburger-Hamilton staging for chick development); V, ventral; VC, ventral commissure.

lack of newly identified cues in the past decade has suggested that the major classes of growth and guidance cues have now been identified.

The commissures in the rodent spinal cord are one of the most prominent model systems to study axon growth and guidance. In a search for the differentially expressed genes in the dorsal spinal cord of mouse embryos, we identified a gene encoding the secreted protein cerebellin-1 (Cbln1). Cbln1 is released from cerebellar parallel fibers and has previously been characterized as a synaptic organizer by forming the synapse-spanning tripartite complex Nrxn-Cbln1-GluD2 (Nrxn, neurexin; GluD2, the ionotropic glutamate receptor family member delta-2) [19,20]. However, whether Cbln1 is expressed and plays roles in earlier nervous system development is unknown.

Here, we found that Cbln1 is expressed both in the dorsal commissural neurons (DCN) and in the floor plate (FP) of the embryonic mouse spinal cord. We generated DCN- and FP-specific *Cbln1* conditional knockout (cKO) mice that demonstrated that the cell-autonomous and non-cell-autonomous Cbln1 from DCNs and FP regulate commissural axon (CA) growth and guidance, respectively. The dual roles of Cbln1 are mediated by its receptor, neurexin-2. Interestingly, the functions and mechanisms of Cbln1 in regulating axon growth and guidance were replicated in the developing cerebellar granule axon growth and the embryonic retinal ganglion cell axon guidance, respectively. Together, our findings reveal a general role for Cbln1 in regulating axon growth and guidance during early nervous system development prior to synapse formation.

## Results

### Cbln1 is expressed in both dorsal commissural neurons and floor plate in the developing mouse spinal cord

To identify the differentially expressed genes over developmental time in the mouse embryonic dorsal spinal cord, we genetically labeled embryonic dorsal spinal neurons with eGFP by crossing *Wnt1-cre* [21,22] with *Rosa26mTmG* [23] mice (S1A Fig). Mouse embryonic E10.5, E11.5, and E12.5 spinal cords were dissected, and dorsal spinal neurons were collected (S1B Fig). Then, GFP$^+$ dorsal spinal neurons were purified by fluorescence-activated cell sorting (FACS), and the differentially expressed genes (DEGs) during the developmental stages were identified by the expression profiling analysis using microarray analysis (S1C Fig and S1 Table). We carried out in situ hybridization to further explore the expression patterns of the candidate DEGs in the developing spinal cord. Among the candidates, *Cbln1* was notable due to its expression pattern. *Cbln1* has strong signals in FP and weak signals in DCNs at E10.5 (Fig 1A). At E11.5 and E12.5, the expression of *Cbln1* increases in DCNs (notice DCNs migrate ventrally and medially at E12.5), maintains a high level in FP, and also appears in subpopulations of motor neurons (Fig 1A).

To further explore the expression patterns of Cbln1 protein and confirm its expression sites, we carried out immunofluorescence (IF) using a Cbln1 antibody [24]. Co-immunostaining of Cbln1 with Lhx2, a dI1 DCN marker [25], confirmed the expression of Cbln1 in the dI1 DCNs of developing spinal cords from E10.5 to E12.5 (Fig 1B–1D). Expression of Cbln1 was also detected in dI2-4 DCN subpopulations (S1D and S1E Fig). Expression of Cbln1 in FP was also confirmed by co-immunostaining with the FP marker Alcam (Fig 1B–1D). To validate the specificity of Cbln1 expression in FP, we used a spinal FP-deficient model, *Gli2* knockout (KO) mouse [26]. As shown in S1F Fig, IF signal of Cbln1 in Alcam-marked FP was gone in *Gli2* KO. These results revealed an interesting expression pattern for Cbln1 that is expressed both in the DCNs and in the intermediate target for DCNs, the FP.

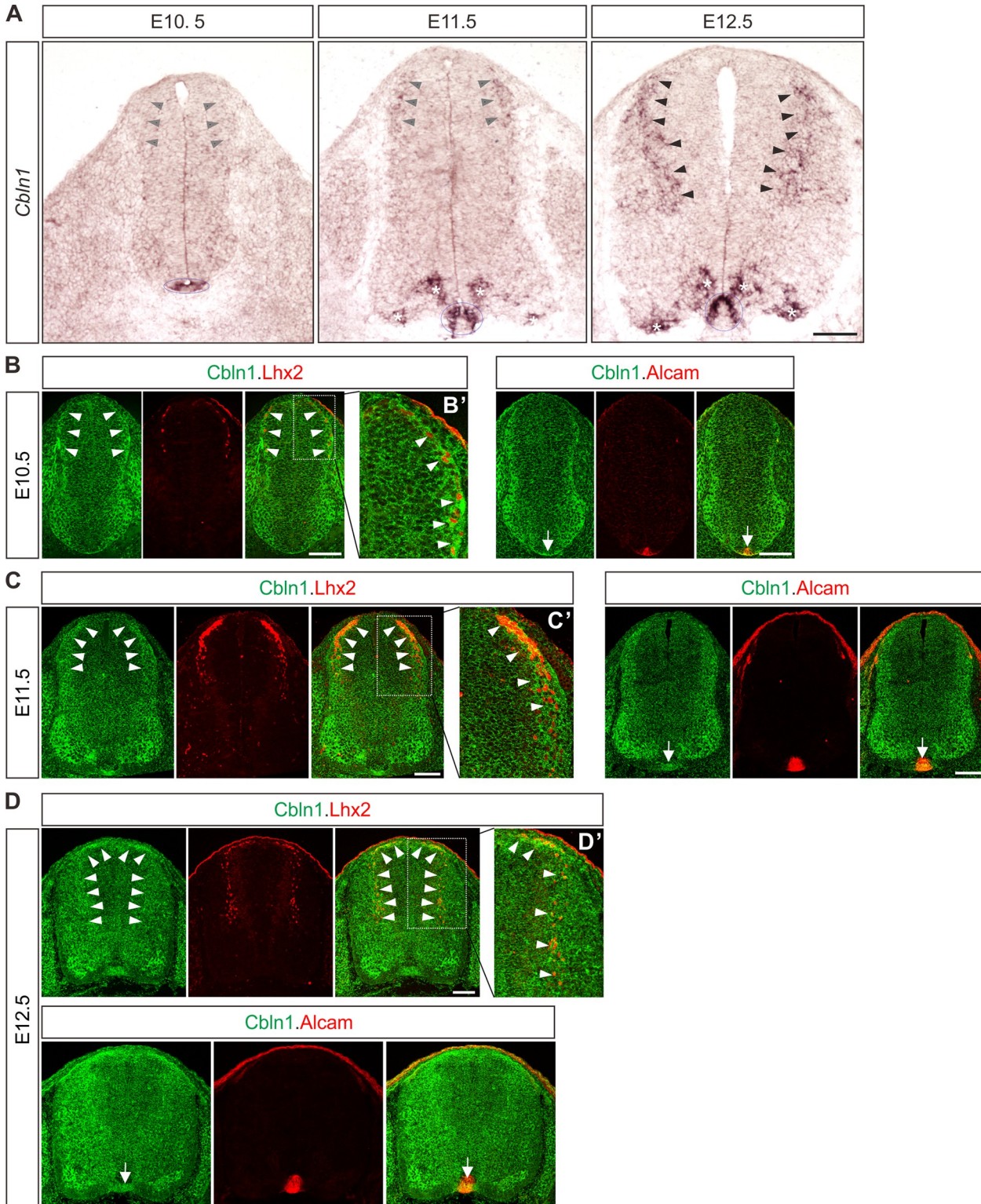

**Fig 1. Expression patterns of *Cbln1* and Cbln1 in the developing mouse spinal cord.** (A) In situ hybridization was carried out using a DIG-labelled RNA probe against *Cbln1* in spinal cord cross-sections during E10.5~E12.5. Arrowheads, circled areas, and asterisks indicate the expression of *Cbln1* in the DCNs, the FP, and motor neurons, respectively. Scale bar, 100 μm. (**B–D**) Co-immunostaining of Cbln1 with Lhx2 or Alcam in spinal cord cross-sections during E10.5 (B), E11.5 (C), and E12.5 (D) showed expression of Cbln1 in the DCN neurons (white arrowheads) and FP (arrows). Higher magnification insets were shown to highlight the expression of Cbln1 in Lhx2+ dI1 neurons (B'–D'). Scale bars, 100 μm. Cbln1, cerebellin 1; DIG, digoxigenin; DCNs, dorsal commissural neurons; dI1, dorsal interneuron 1.

## Cell-autonomous Cbln1 in the dorsal commissural neurons is both required and sufficient to stimulate commissural axon growth in vivo

Next, we wanted to explore the roles of Cbln1 expressed in DCNs and FP, separately. In order to specifically ablate *Cbln1* from these tissues, we generated cKO of *Cbln1* using tissue-specific *Cre* lines (Fig 2A). We used *Wnt1-Cre* line to specifically ablate *Cbln1* from spinal DCNs, without affecting *Cbln1* expression in other parts of spinal cord (Fig 2B). *Cbln*1 cKO in spinal DCNs does not disturb neurogenesis of these neurons, as indicated by normal numbers, distribution and patterning of dl1-4 neurons in the developing spinal cord (S2A–S2H Fig). We continued to check CA growth in DCN-specific *Cbln1* cKO. We prepared open-books of developing spinal cords and used Robo3 immunostaining to label CAs. Robo3 selectively marks CAs as they navigate to and across the FP [27]. As shown in Fig 2C–2E, both lengths and numbers of CAs were decreased in *Cbln1* cKO embryos compared with their littermate controls. These data suggest that Cbln1 in the DCNs is required for their own commissural axon growth.

To further test whether Cbln1 is sufficient to stimulate CA growth in vivo, we used a model of chick neural tube. Chick *Cbln1* (*cCbln1*) is expressed in the DCNs of developing chick neural tube, as is the case with mouse *Cbln1*, but is not detected in the FP of chick embryonic spinal cord (Fig 2F). We made a DCN-specific overexpression plasmid, pMath1-eGFP-IRES-MCS, by modifying a DCN-specific knockdown plasmid pMath1-eGFP-miRNA [28]. Unilateral DCN-specific overexpression of *cCbln1* by in ovo electroporation of pMath1-eGFP-IRES-cCbln1 enhanced chick CA growth compared with control plasmid without changing commissural neuron numbers (Fig 2G–2I). These data suggest that Cbln1 is sufficient to stimulate CA growth.

In the DCN-specific *Cbln1* cKO embryonic spinal cord, other cell-autonomous growth promoting molecules or non-cell-autonomous guidance attractants/repellents are not affected. So we continued to check whether the CA growth could eventually reach the FP. As shown in S2I Fig, almost all axons reached the floor plate at E12, and there was no difference between *Cbln1* cKO and control for commissural axon length or number at this stage (S2J and S2K Fig). In an open book DiI assay performed at E12, the crossing or post-crossing behaviors of CAs in DCN-specific *Cbln1* cKO embryos were also similar to those of controls (S2L Fig).

## Cbln1 is secreted from commissural axon growth cones and stimulates commissural axon growth in an autocrine manner

Next, we continued to elucidate the mechanisms for the cell-autonomous functions of Cbln1. We hypothesized that Cbln1 was secreted from the DCNs and then acted to stimulate CA growth in an autocrine manner. To test this, we cultured DCN explants from E10.5 (a stage when most CAs have not projected to the midline yet and are called pre-crossing axons) mouse spinal cords and used Tag1 immunostaining to visualize pre-crossing commissural axons. Tag1 has been widely used as a marker for pre-crossing CAs [29,30]. Compared with control embryonic DCN explants, the CA growth of Wnt1-Cre-mediated *Cbln1* cKO DCNs was significantly inhibited, indicated by decreased axon numbers and reduced axon lengths (Fig 3A–3C), which is consistent with in vivo results for DCN-specific *Cbln1* cKO (Fig 2B–2E). These axon growth defects were efficiently rescued by adding a recombinant human Cbln1 protein (rhCbln1) to the cultures (Fig 3A–3C). These data suggest that the cell-autonomous Cbln1 regulates CA growth in an autocrine manner.

We next asked whether Cbln1-induced axonal growth works locally in CAs and growth cones. Immunofluorescence of DCN neuron culture using a Cbln1 antibody detected robust Cbln1 IF signals in CAs and growth cones (Fig 3D). To confirm the specificity of these axonal

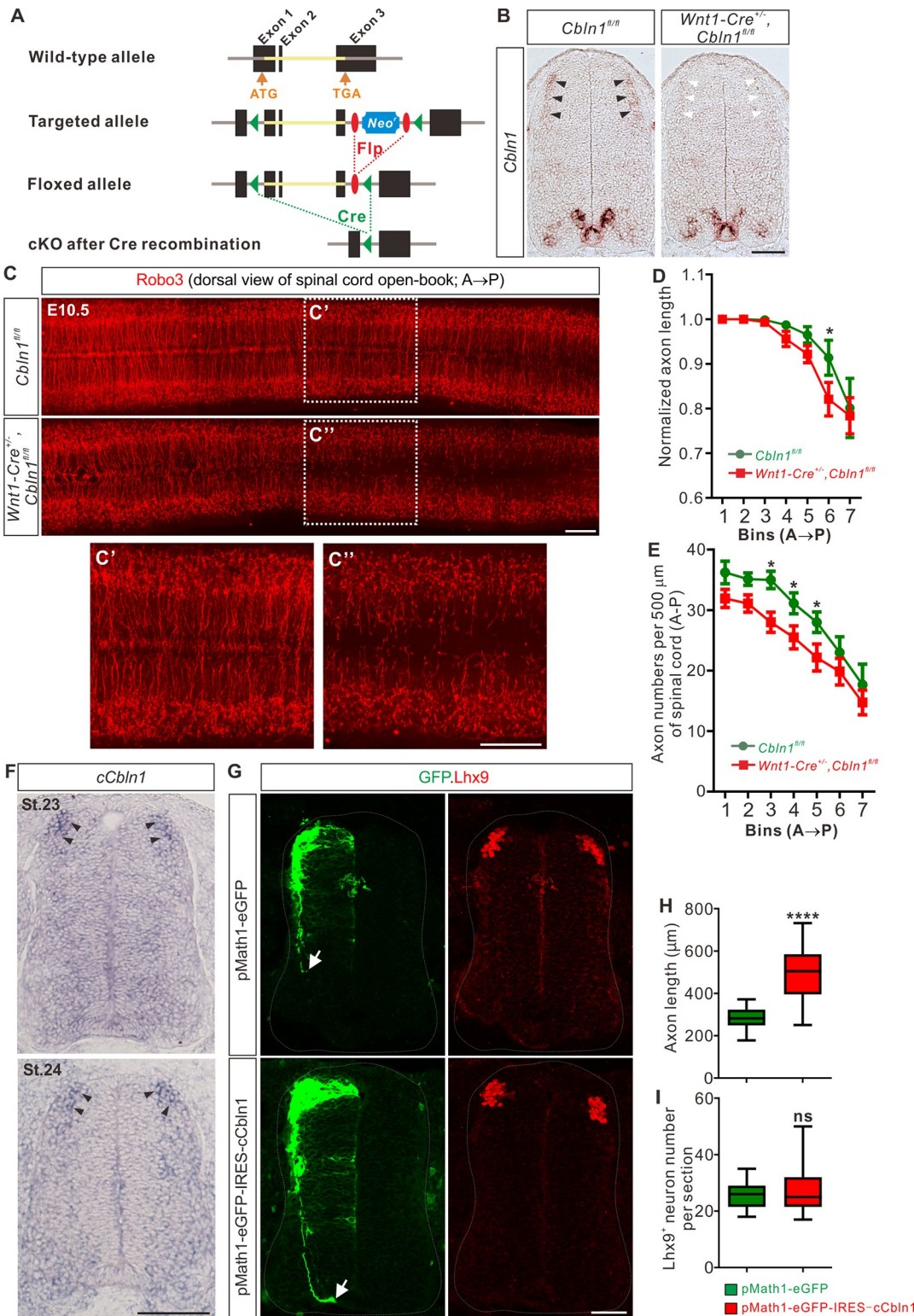

**Fig 2. Cell-autonomous Cbln1 is both required and sufficient to stimulate commissural axon growth in vivo.** (**A**) Schematic drawings showing the generation of *Cbln1* cKO. The coding sequence of *Cbln1* is deleted after Cre-mediated recombination. (**B**) Specific depletion of *Cbln1* in the dorsal spinal cord of *Wnt1-Cre*$^{+/-}$,*Cbln1*$^{fl/fl}$ cKO mouse embryos. In situ hybridization of E11.5 spinal cord sections using RNA probes against *Cbln1* confirmed specific ablation of *Cbln1* from the DCNs. Black arrowheads indicate *Cbln1* expression in control DCNs and white arrowheads highlight the missing *Cbln1*

expression in cKO DCNs. Expression of *Cbln1* in FP (in red dotted circles) and other parts are not affected in DCN-specific *Cbln1* cKO spinal cords. Scale bar, 100 μm. (**C**) DCN-specific *Cbln1* cKO caused dramatic CA growth defects in vivo. Commissural axons were marked by Robo3 immunostaining in spinal cord open-books at E10.5. The lengths of CAs are much shorter and the numbers of CAs are much fewer in *Cbln1* cKO spinal cords compared with their littermate controls. Notice that these differences are more obvious in posterior ends of spinal cords. The dotted boxed areas were shown in the higher magnification insets (C' and C"). A, anterior; P, posterior. Scale bar, 200 μm. (**D** and **E**) Quantification of commissural axon lengths and numbers in (C). The spinal cords were divided to bins (500 μm) along the anterior-posterior (A→P) direction, and the lengths (D) and numbers (E) of CAs in each bin were quantified. All data are mean ± SEM: *Cbln1*$^{fl/fl}$ ($n = 9$ embryos) vs. *Wnt1-Cre*$^{+/-}$,*Cbln1*$^{fl/fl}$ ($n = 12$ embryos); $^*p = 0.014$ for Bin 6 in D; $^*p = 0.015$ for Bin 3 in E; $^*p = 0.049$ for Bin 4 in E; $^*p = 0.041$ for Bin 5 in E; by multiple *t* tests. (**F**) In situ hybridization of *cCbln1* in spinal cord cross-sections of St.23-24 chick embryos. Arrowheads indicate the expression of *cCbln1* in DCNs. Scale bar, 50 μm. (**G**) Unilateral DCN-specific overexpression of *cCbln1* by in ovo electroporation of pMath1-eGFP-IRES-cCbln1 enhanced commissural axon growth in chick neural tubes. Lhx9 marks dI1 DCNs and eGFP marks electroporated DCNs and their axons. The arrows point commissural axon terminals. Shown are the representative images from 10 chick embryos with pMath1-eGFP-IRES-cCbln1 and 8 embryos with control plasmid. Scale bar, 50 μm. (**H** and **I**) Quantification of commissural axon length and Lhx9$^+$ neuron numbers in (G). All data are represented as box and whisker plots: for H, pMath1-eGFP-IRES-cCbln1 ($n = 35$ sections) vs. pMath1-eGFP ($n = 31$ sections), $^{****}p = 5.56 \times 10^{-12}$; for I, pMath1-eGFP-IRES-cCbln1 ($n = 37$ sections) vs. pMath1-eGFP ($n = 29$ sections), $p = 0.32$, ns, not significant; by unpaired Student *t* test. The data underlying all the graphs shown in the figure are included in S1 Data. CA, commissural axon; Cbln1, cerebellin 1; Cre, Cre recombinase; DCNs, dorsal commissural neurons; cKO, conditional knockout; A→P, anterior to posterior; SEM, standard error of the mean; St., stage (Hamburger-Hamilton staging for chick development).

Cbln1 IF signals, we generated lentiviral sh*Cbln1* that led to dramatic knockdown of *Cbln1* in cultured neurons (S3A Fig). The Cbln1 IF signals in CAs and growth cones were largely lost after knockdown of Cbln1 (Fig 3D and 3E), indicating that Cbln1 is present in CAs and growth cones.

We continued to explore how Cbln1 is released from CAs. It is reported that Cbln1 co-localizes with the lysosomal enzymes cathepsin B and D in the adult mouse brain [31,32], indicating the lysosome may regulate Cbln1 secretion in CAs. To test this, we applied different lysosome inhibitors to the DCN cultures and checked their effects on Cbln1 secretion from CAs. Glycyl-L-phenylalanine 2-naphthylamide (GPN) can be specifically cleaved by cathepsin C, which leads to targeted disruption of the lysosomal membrane [32,33]. Bafilomycin A1 (BafA$_1$) blocks lysosomal functions through working as a specific inhibitor of vacuolar H$^+$-ATPase [34]. Treatment of DCN cultures with GPN or BafA$_1$, followed by an IF protocol to detect surface Cbln1 by leaving out the permeabilization steps, showed loss of Cbln1 IF signals on the CA surface (Fig 3F and 3G, S3B and S3C Fig), suggesting that Cbln1 is released from lysosomes in CAs and growth cones. Blocking Cbln1 secretion by GPN inhibited CA growth (Fig 3H), further supporting a model that Cbln1 is released from and works back on CA and growth cones to stimulate axon growth.

## Non-cell-autonomous Cbln1 from the floor plate regulates commissural axon guidance

The facts that the secreted Cbln1 works extrinsically and that it is expressed in the FP during CA growth to the midline suggest that Cbln1 from FP might regulate CA guidance. To test this idea, we first prepared COS7 cell lines stably expressing mouse Cbln1. High levels of Cbln1 were detected in the culture media, indicating the overexpressed Cbln1 was secreted from COS7 cells (S4A Fig). We then co-cultured the dorsal spinal cord explants from E11 mouse embryos with COS7 cell aggregates expressing Cbln1 tagged with FLAG and ZsGreen or ZsGreen alone in collagen gels (Fig 4A). The dorsal spinal cord explants growing with Cbln1-expressing COS7 cell aggregates had significantly longer axons than the control (Fig 4A and 4B). More importantly, the growth of commissural axons was attracted toward Cbln1-expressing COS7 cell aggregates, indicated by the higher axon number ratios (proximal/distal)

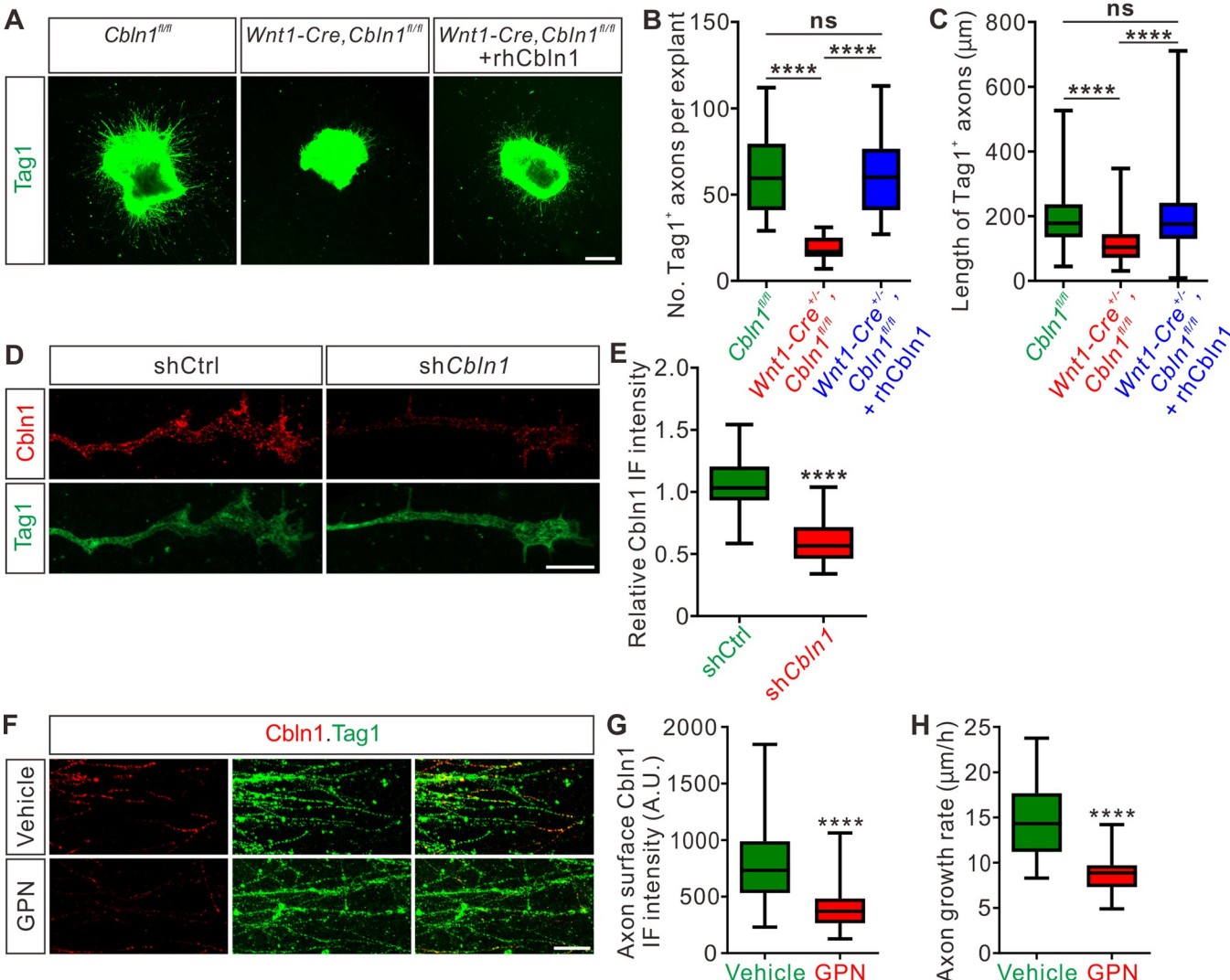

**Fig 3. Cbln1 is secreted from commissural axon growth cones and stimulates commissural axon growth in an autocrine manner.** (**A**) Extrinsic Cbln1 could rescue CA growth defects caused by cell-autonomous ablation of *Cbln1* in the DCNs. DCN explants dissected from E10.5 mouse embryos were cultured in vitro and CA length was monitored by immunostaining of Tag1, a CA marker. Compared with *Cbln1^{fl/fl}*, DCN explants of *Wnt1-Cre^{+/-},Cbln1^{fl/fl}* embryos showed significant CA growth defects. These defects were rescued by adding the recombinant human Cbln1 protein (rhCbln1) to the cultures. Scale bar, 200 μm. (**B** and **C**) Quantification of Tag1$^+$ commissural axon numbers and lengths in (A). Data are represented as box and whisker plots. For B, $^{****}p = 1.69 \times 10^{-6}$, *Cbln1^{fl/fl}* (*n* = 14 explants) vs. *Wnt1-Cre^{+/-},Cbln1^{fl/fl}* (*n* = 14 explants); $^{****}p = 6.23 \times 10^{-7}$, *Wnt1-Cre^{+/-},Cbln1^{fl/fl}* vs. *Wnt1-Cre^{+/-},Cbln1^{fl/fl}* + rhCbln1 (*n* = 16 explants); ns, not significant (*p* = 0.99), *Cbln1^{fl/fl}* vs. *Wnt1-Cre^{+/-},Cbln1^{fl/fl}* + rhCbln1. For C, $^{****}p = 6.61 \times 10^{-11}$, *Cbln1^{fl/fl}* (*n* = 876 axons) vs. *Wnt1-Cre^{+/-},Cbln1^{fl/fl}* (*n* = 274 axons); $^{****}p = 6.61 \times 10^{-11}$, *Wnt1-Cre^{+/-},Cbln1^{fl/fl}* vs. *Wnt1-Cre^{+/-},Cbln1^{fl/fl}* + rhCbln1 (*n* = 1,013 axons); ns, not significant (*p* = 0.91), *Cbln1^{fl/fl}* vs. *Wnt1-Cre^{+/-},Cbln1^{fl/fl}* + rhCbln1. By 1-way ANOVA followed by Tukey's multiple comparison test. (**D**) Robust Cbln1 IF signals were detected in CAs and growth cones. Dissociated DCN neurons from E11 mouse embryos were cultured in vitro and Cbln1 IF signals were imaged after lentiviral shRNA infection. Loss of Cbln1 IF signals after sh*Cbln1* infection indicated the specificity of Cbln1 IF signals in CAs and growth cones. Scale bar, 10 μm. (**E**) Quantification of axonal Cbln1 IF signals in (D). Data are represented as box and whisker plots: shCtrl (*n* = 55 axons) vs. sh*Cbln1* (*n* = 68 axons), $^{****}p = 5.65 \times 10^{-31}$, by unpaired Student *t* test. (**F**) Cbln1 is exocytosed from CAs via lysosomes. Robust Cbln1 IF signals were detected on the CA surface of cultured DCN explants and were eliminated after blocking exocytosis with GPN treatment for 10 min. Scale bar, 50 μm. (**G**) Quantification of axon surface Cbln1 IF signals in (F). Data are represented as box and whisker plots: Vehicle (*n* = 140 axons) vs. GPN (*n* = 126 axons), $^{****}p = 1.94 \times 10^{-26}$, by unpaired Student *t* test. (**H**) Blocking Cbln1 exocytosis in CA with GPN for 7 h inhibited CA growth. Data are represented as box and whisker plots: Vehicle (*n* = 75 axons) vs. GPN (*n* = 51 axons), $^{****}p = 2.15 \times 10^{-20}$, by unpaired Student *t* test. The data underlying all the graphs shown in the figure are included in S1 Data. A.U., arbitrary unit; CA, commissural axon; Cbln1, cerebellin 1; DCN, dorsal commissural neuron; GPN, Glycyl-L-phenylalanine 2-naphthylamide; IF, immunofluorescence; rhCbln1, recombinant human Cbln1 protein; shCtrl, control shRNA; sh*Cbln1*, shRNA against *Cbln1*.

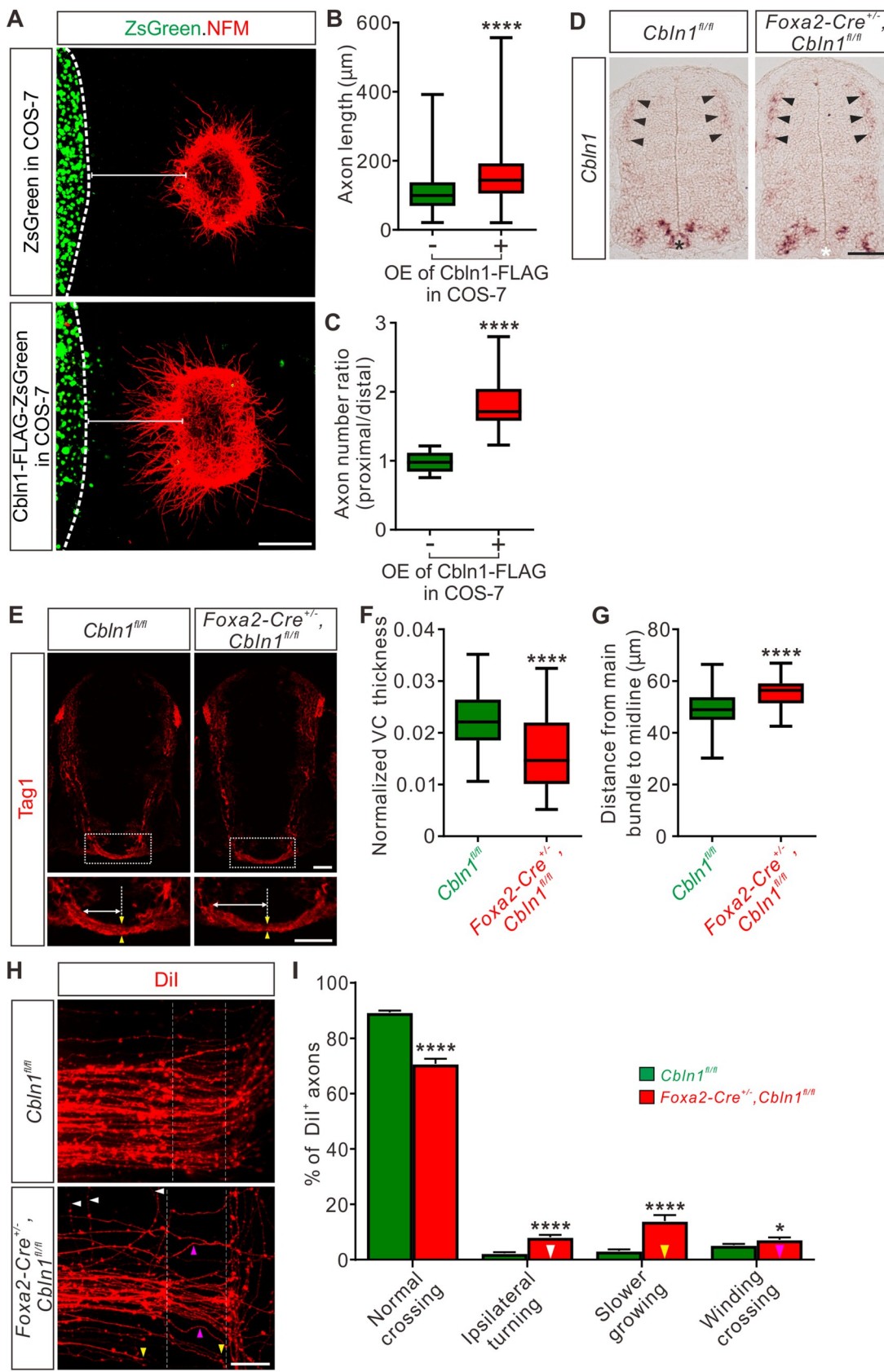

**Fig 4. Non-cell-autonomous Cbln1 from the floor plate regulates commissural axon guidance.** (**A**) Co-culture of DCN explants from E11 mouse spinal cords with COS7 cell aggregates expressing Cbln1-FLAG with ZsGreen or ZsGreen alone. Commissural axons were visualized with NFM immunostaining. Cbln1 expression attracted CA turning toward cell aggregates and also enhanced axon growth. Scale bar, 200 μm. (**B** and **C**) Quantification of CA growth and turning in (A) by measuring the axon length (B) and the axon number ratio (proximal/distal) (C). All data are represented as box and whisker plots: for B, Ctrl ($n$ = 1,336 axons) vs. OE ($n$ = 1,051 axons), $^{****}p = 1.93 \times 10^{-70}$; for C, Ctrl ($n$ = 16 explants) vs. OE ($n$ = 14 explants), $^{****}p = 1.97 \times 10^{-8}$; by unpaired Student $t$ test. (**D**) Specific ablation of *Cbln1* in the floor plate of *Foxa2-Cre$^{+/-}$,Cbln1$^{fl/fl}$* cKO mouse embryos was confirmed by in situ hybridization of E11.5 spinal cord sections. Expression of *Cbln1* in the floor plate was completely lost in the cKO spinal cord (white asterisk) compared with the control embryos (black asterisk). Black arrowheads indicate the unchanged *Cbln1* expression in DCNs of both genotypes. Scale bar, 100 μm. (**E**) The axon guidance defects of pre-crossing commissural axons were observed by Tag1 immunostaining in floor plate-specific *Cbln1* cKO and control embryos at E11.5. Higher magnification views of the floor plate region in the white dotted boxes are also shown (bottom). The pair of yellow arrowheads denotes the thickness of the VC. The double-arrowed line measures the distance between the point of intersection (of the main pre-crossing CA bundle with the ventral edge of spinal cord) and the midline (indicated by the dotted line). Scale bars, 50 μm. (**F** and **G**) Quantification of the VC thickness and the distance from the main bundle intersection point to the midline. The VC thickness was normalized to the height (dorsal to ventral) of spinal cord. All data are represented as box and whisker plots: *Cbln1$^{fl/fl}$* ($n$ = 62 sections) vs. *Foxa2-Cre$^{+/-}$,Cbln1$^{fl/fl}$* ($n$ = 60 sections), $^{****}p = 1.69 \times 10^{-6}$ for F, $^{****}p = 1.07 \times 10^{-7}$ for G, by unpaired Student $t$ test. (**H**) DiI labeling of E11.5 spinal cord open-books traced commissural axon guidance behaviors during midline crossing. The region between 2 white dotted lines indicates the floor plate. The white, yellow, and purple arrowheads indicate the CAs with aberrant behaviors such as ipsilateral turning, slower growing or winding crossing, respectively. Scale bar, 50 μm. (**I**) Quantification of the percentages of CAs with different guidance behaviors. All data are mean ± SEM and represented as histogram: *Cbln1$^{fl/fl}$* ($n$ = 45 DiI injections) vs. *Foxa2-Cre$^{+/-}$,Cbln1$^{fl/fl}$* ($n$ = 32 DiI injections), $^{****}p = 3.36 \times 10^{-17}$ for normal crossing, $^{****}p = 8.86 \times 10^{-10}$ for ipsilateral turning, $^{****}p = 6.14 \times 10^{-7}$ for slower growing, $^{*}p = 0.033$ for winding crossing, by unpaired Student $t$ test. The data underlying all the graphs shown in the figure are included in S1 Data. CA, commissural axon; Cbln1, cerebellin 1; cKO, conditional knockout; Ctrl, control; DCNs, dorsal commissural neurons; NFM, neurofilament; OE, overexpression; SEM, standard error of the mean; VC, ventral commissure.

compared with the control cell aggregates expressing ZsGreen alone (Fig 4A and 4C). These results suggest that the non-cell-autonomous Cbln1 functions as an attractive axon guidance molecule.

To assess the in vivo functions of the non-cell-autonomous Cbln1, we generated FP-specific *Cbln1* cKO mice. We utilized *Foxa2-Cre$^{ERT}$* line that has been used to induce Cre recombinase expression specifically in floor plate cells in response to tamoxifen (TM) treatment [35,36]. *Cbln1* expression was specifically ablated from the FP in these cKO embryos, without affecting its expression in other parts of spinal cord including the DCNs (Fig 4D). The neural patterning or neurogenesis was not disturbed by ablation of Cbln1 from the FP (S4B–S4E Fig). However, examination of commissural axon trajectories using Tag1 immunostaining in E11.5 spinal cords revealed significant axon guidance defects in the midline and ventral spinal cord. First, the thickness of the ventral commissure (VC) was significantly reduced in the FP-specific *Cbln1* cKO embryos compared with their littermate controls (Fig 4E and 4F). Second, the intersection of the main CA bundle with the ventral commissural funiculus was shifted laterally in the FP-specific *Cbln1* cKO embryos compared with their littermate controls (Fig 4E). The distances between the point of intersection and the midline were quantified, showing a significant increase in the FP-specific *Cbln1* cKO embryos (Fig 4G). These phenotypes were also evident by NFM immunostaining (S4F–S4H Fig). These axon guidance defects suggest that Cbln1 from the FP indeed works as an axon guidance cue in the developing spinal cord.

To observe more clearly the CA guidance behaviors in the FP-specific *Cbln1* cKO embryos, we performed DiI labeling of DCNs in the open-book spinal cords at E11.5. As shown in Fig 4H and 4I, there was a significant decrease of the number of normal crossing CAs in the *Cbln1* cKO. Meanwhile, the numbers of commissural axons showing guidance defects such as ipsilateral turning, slower growing or winding crossing were significantly increased in *Cbln1* cKO embryos compared with their littermate controls (Fig 4H and 4I).

All these data support the idea that the non-cell-autonomous Cbln1 derived from the FP works as an axon guidance cue for CAs in the developing spinal cord.

## Nrxn2 is expressed in the developing dorsal commissural neurons and axons, and mediates Cbln1-induced axon growth and guidance as its receptor

Previously, Cbln1 was shown to work as a synaptic organizer for the cerebellar excitatory PF-PC (PF, parallel fibers; PC, Purkinje cells) synapses by binding to its presynaptic receptor, neurexin (Nrxn) and its postsynaptic receptor, glutamate receptor delta 2 (GluD2) [37,38]. In addition, trans-synaptic signaling through Nrxn-Cbln-GluD1 has also been shown to mediate the inhibitory synapse formation in cortical neurons [39,40]. Here, we wondered whether Nrxn, GluD1, and/or GluD2 work as receptors for Cbln1 to mediate its regulation of CA growth and guidance in the developing spinal cord. To test this, we first checked if Nrxn, GluD1, and GluD2 are expressed in developing DCNs or not. *GluD1* or *GluD2* mRNA was not detected in E11.5 spinal cords (S5A Fig). There are 3 *Nrxn* genes in the mammalian genome, each of them encoding 2 major protein isoforms, α-neurexin and β-neurexin [41]. Although *Nrxn1*, *2*, and *3* mRNAs were all detected in E11.5 spinal cords, only *Nrxn2* mRNA was found to be expressed in DCNs (Fig 5A). We further detected both *Nrxn2α* and *Nrxn2β* mRNAs in DCNs using the isoform-specific probes (Fig 5B). Immunostaining using an Nrxn2 antibody detected robust Nrxn2 IF signals in commissural axons and growth cones (Fig 5C and 5D), making it possible that Nrxn2 works as the receptor for Cbln1 in developing CAs.

Indeed, CA lengths were significantly decreased after knocking down either pan-Nrxn2 or Nrxn2α, Nrxn2β separately (S5B–S5D Fig, Fig 5E and 5F). We continued to observe that axonal surface Cbln1 signals were decreased after Nrxn2 knockdown in culture DCN neurons (S5E and S5F Fig). These data suggest that Nrxn2 mediates cell-autonomous-Cbln1-induced CA growth. We continued to test whether Nrxn2 also mediates the non-cell-autonomous function of Cbln1 to attract CA turning. As shown in Fig 5G, Cbln1-expressing cell aggregates failed to attract commissural axons of DCN neurons that were knocked down of either pan-Nrxn2 or Nrxn2α, Nrxn2β separately. These data suggest that Nrxn2 (both Nrxn2α and Nrxn2β) works as the receptor for Cbln1 to mediate its cell-autonomous function in axon growth and non-cell-autonomous function in axon guidance of commissural neurons in the developing spinal cord.

In summary, these data and findings support the following working model for Cbln1 in the developing spinal cord (S5G and S5H Fig). In the pre-crossing CAs, Cbln1 is expressed cell-autonomously by the DCN and axons. Commissural axon growth cone-secreted Cbln1 binds to Nrxn2 receptors in commissural axons and growth cones to stimulate commissural axon growth in an autocrine manner (S5G Fig). In the DCN-specific *Cbln1* cKO embryos, CA growth is reduced compared with their littermate controls (S5G Fig). When commissural axons approach the midline, the FP-derived Cbln1 attracts CAs to the midline that is also mediate by Nrxn2 receptors (S5H Fig). In the FP-specific *Cbln1* cKO embryos, commissural axon guidance in the midline crossing is impaired, resulting in a U-shaped and thinner ventral commissure compared with the V-shaped and thick ventral commissures in the littermate control embryos (S5H Fig).

## Cell-autonomous Cbln1 from cerebellar granular cells is required for parallel fiber growth

We wondered whether the function of Cbln1 to regulate axon development is a general mechanism which also works in other brain regions during development. The studies on Cbln1 so far have been focused on its functions as a synaptic organizer in cerebellum. Whether Cbln1 is expressed and exerts its functions at earlier stages of cerebellar development remains unexplored. We first checked expression of Cbln1 in earlier cerebellar development. As shown in Fig 6A, high and specific *Cbln1* expression was detected in the P4-P8 cerebellar granule cells in

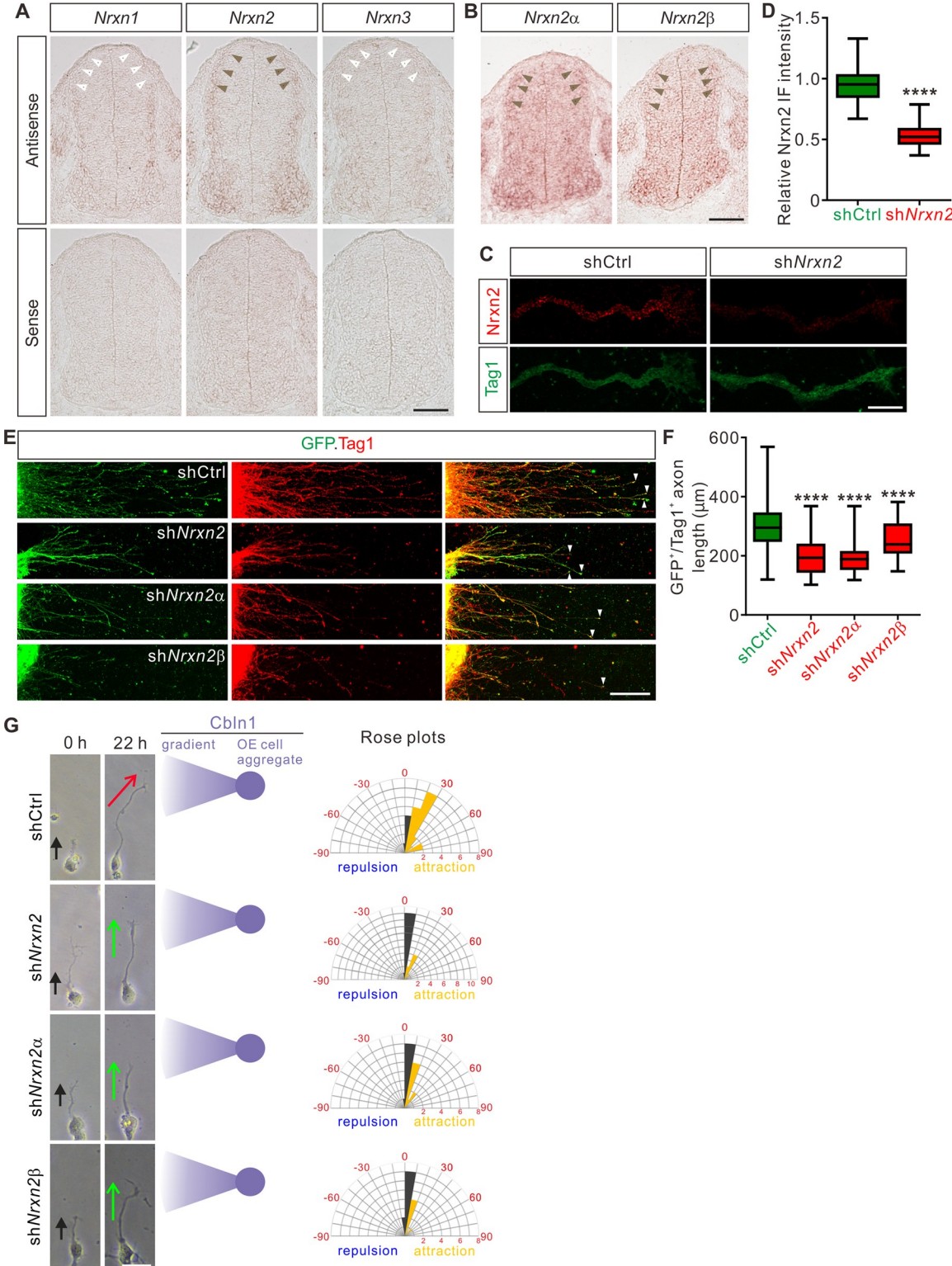

**Fig 5. Nrxn2 mediates Cbln1-induced commissural axon growth and guidance as its receptor.** (**A** and **B**) *Nrxn2*, *Nrxn2α*, and *Nrxn2β* mRNAs were detected in E11.5 spinal cord cross-sections by in situ hybridization. Brown and white arrowheads indicate the expression (*Nrxn2* in A, *Nrxn2α* and *Nrxn2β* in B) or absence (*Nrxn1* and *Nrxn3* in A) of the corresponding mRNAs in DCNs, respectively. Scale bars, 100 μm (A, B). (**C**) Robust Nrxn2 IF signals were detected in the commissural axons and growth cones. Dissociated DCN neurons from E11 mouse embryos were cultured in vitro and Nrxn2 IF signals were imaged after lentiviral shRNA

infection. Loss of Nrxn2 IF signals after sh*Nrxn2* infection indicated the specificity of Nrxn2 IF signals in the commissural axons and growth cones. Scale bar, 10 μm. (**D**) Quantification of axonal Nrxn2 IF signals in (C). Data are represented as box and whisker plots: shCtrl ($n = 69$ axons) vs. sh*Nrxn2* ($n = 65$ axons), ****$p = 3.75 \times 10^{-39}$, by unpaired Student $t$ test. (**E**) Knockdown of Nrxn2, Nrxn2α, or Nrxn2β in DCNs inhibited commissural axon growth. Axons of DCN neurons that were infected by shRNA against *Nrxn2* were marked by both GFP reporter and Tag1 IF. White arrowheads indicate the axon terminals. Scale bar, 100 μm. (**F**) Lengths of GFP$^+$/Tag1$^+$ commissural axons in (E) were measured and analyzed. Data are represented as box and whisker plots: shCtrl ($n = 63$ axons) vs. sh*Nrxn2* ($n = 53$ axons), ****$p = 1.41 \times 10^{-15}$; shCtrl vs. sh*Nrxn2α* ($n = 48$ axons), ****$p = 4.30 \times 10^{-18}$; shCtrl vs. sh*Nrxn2β* ($n = 48$ axons), ****$p = 6.27 \times 10^{-5}$; by 1-way ANOVA followed by Tukey's multiple comparison test. (**G**) Knockdown of Nrxn2, Nrxn2α, or Nrxn2β in DCNs disturbed commissural axon turning toward Cbln1-expressing cell aggregates. Dissociated DCN neurons from E11 mouse spinal cords were infected with shRNAs, and co-cultured with COS7 cell aggregates expressing Cbln1. Commissural axons were imaged at 2 time points (0 and 22 h). Commissural axon turning angles toward the Cbln1-OE cell aggregates and gradients were measured between the colored arrow (red for shCtrl and green for sh*Nrxn2s*) at 22 h and the black arrow at 0 h. Rose plots of axon turning angles are shown to the right for each condition. Angles were clustered in bins of 10˚, and the number of axons per bin is represented by the radius of each segment. Orange bins indicate attraction, and blue bins indicate repulsion. Scale bar, 20 μm. The data underlying all the graphs shown in the figure are included in S1 Data. Cbln1, cerebellin 1; DCNs, dorsal commissural neurons; h, hours; IF, immunofluorescence; Nrxn, neurexin; OE, overexpression; shCtrl, control shRNA; sh*Nrxn*, shRNA against *neurexin*.

the inner granule layer (IGL) by in situ hybridization. Immunofluorescence using a Cbln1 antibody showed that Cbln1 protein is enriched in the molecular layer (ML) of cerebellum (Fig 6B), suggesting that Cbln1 protein is expressed and secreted by cerebellar granule cell (GC) axons.

Next, we tested the possible roles of Cbln1 in earlier cerebellar development. We generated *Cbln1* cKO in cerebellum using the *Wnt1-cre* line [21,42], which resulted in the efficient knockout of *Cbln1* from GCs (Fig 6C). Axon growth rates of Cbln1-deficient GCs in vitro were significantly decreased compared with control neurons (Fig 6D and 6E), suggesting that the cell-autonomous Cbln1 is required for GC axon growth. Similar to Cbln1 on commissural axons, extrinsic application of the recombinant hCbln1 (rhCbln1) protein to the GC axons could efficiently rescue this axon growth defect (Fig 6D and 6E). These results support a similar model as in the developing spinal cord that Cbln1 secreted from cerebellar GC axons works back to stimulate GC axon growth in the developing cerebellum. Detection of *Nrxn1*, *2*, and *3* expression in GCs at IGL (S6A and S6B Fig) implied that neurexins would mediate the autocrine function of Cbln1 to stimulate GC axon growth in the developing cerebellum as in the spinal cord.

We continued to carefully examine the *Cbln1* cKO cerebella. Immunostaining of the Purkinje cell (PC) marker Calbindin and the granule cell (GC) marker NeuN showed no difference between *Cbln1* cKO and control cerebella at P8 (Fig 6F), suggesting that the neurogenesis of PC and GC in the cerebellum is not impaired. To investigate whether the in vitro regulation of GC axon growth by Cbln1 was recapitulated in vivo, we examined parallel fiber (PF) development in *Cbln1* cKO mice by DiI labeling. Compared with control mice, the DiI-labeled parallel fiber lengths in *Cbln1* cKO mouse pups at P6 were significantly decreased (Fig 6G and 6H), indicating that the parallel fiber growth was impaired in *Cbln1* cKO cerebella.

All these data suggested that cell-autonomous Cbln1 from granule cells is required for parallel fiber growth in the developing cerebellum, just as cell-autonomous Cbln1 from commissural axons stimulates CA growth in the developing spinal cord.

## Non-cell-autonomous Cbln1 regulates axon guidance of retinal ganglion cells in the optic chiasm

The regulation of commissural axon guidance during midline crossing by the non-cell-autonomous Cbln1 from FP of the developing spinal cord inspired us to further test whether Cbln1 regulates axon guidance in other brain midline models. The optic chiasm (OC) is where retinal ganglion cell (RGC) axons from each eye cross the midline. The ipsi- and contra-lateral axon

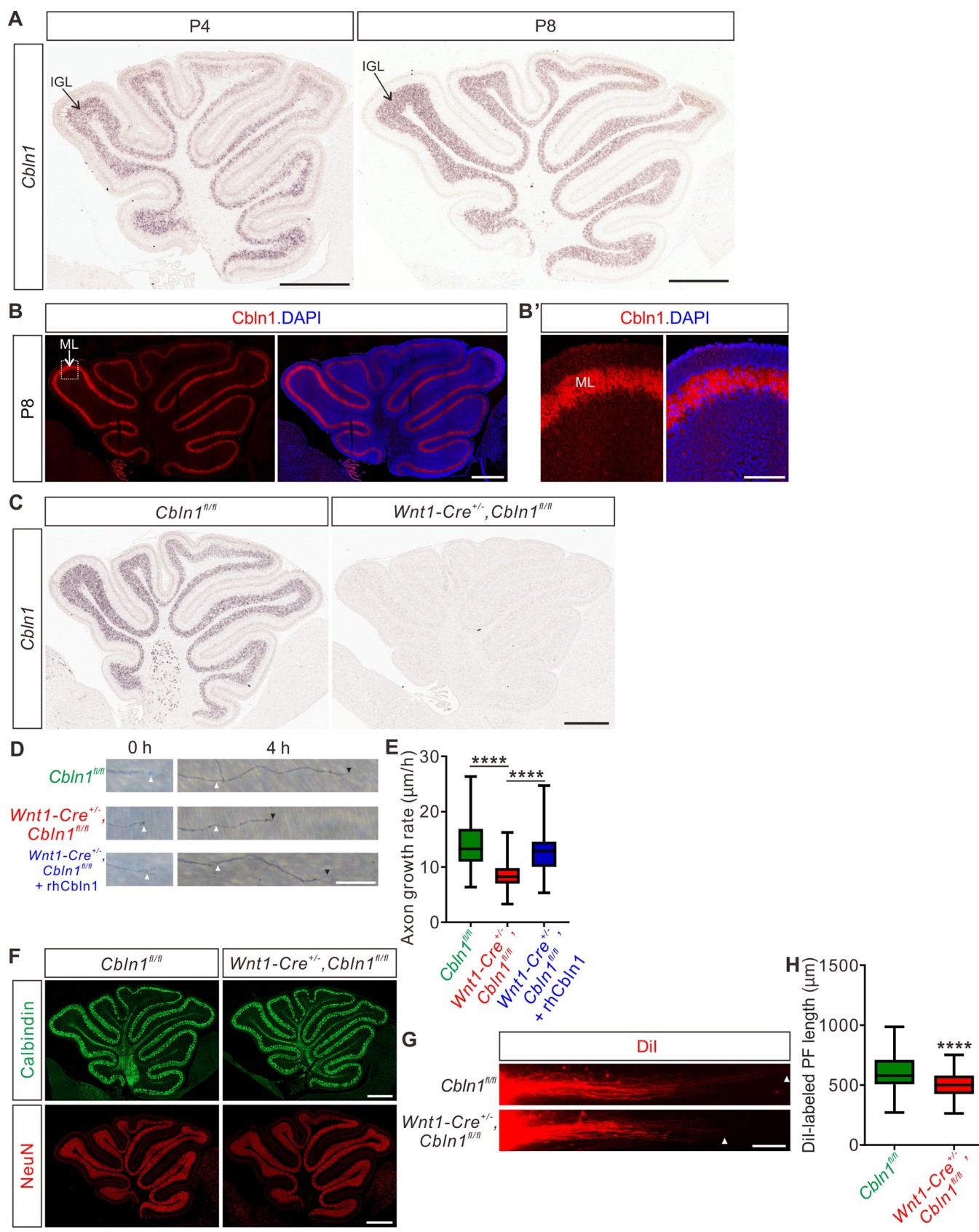

**Fig 6. Cell-autonomous Cbln1 is required for cerebellar granule cell axon growth.** (**A**) In situ hybridization of *Cbln1* in cerebella during P4 and P8. *Cbln1* mRNA is specifically and highly expressed in granule cells, esp. in the IGL. Scale bars, 500 μm. (**B**) High level of Cbln1 protein is detected in the ML of P8 cerebellum, which is expressed and secreted by GC axons. Higher magnification of the boxed area is shown in (B'). Scale bars, 500 μm (B) and 100 μm (B'). (**C**) Ablation of *Cbln1* expression in *Cbln1* cKO mouse cerebella. In situ hybridization of *Cbln1* in P8 cerebellum confirmed the ablation of *Cbln1* from IGL. Scale bar, 500 μm. (**D** and **E**) Extrinsic Cbln1 could rescue GC axon growth defects caused by cell-autonomous ablation of Cbln1 in cerebellar GC neurons. P8 GC neurons were dissected and cultured in vitro. GC axons were imaged at 2 time points (0 and 4 h). The growth rate of GC axons from *Cbln1* cKO cerebella was significantly slower than that of control. This defect was rescued by adding the recombinant human Cbln1 protein (rhCbln1) to the cultures. Quantification data are represented as box and whisker plots (E): *Cbln1*$^{fl/fl}$ ($n$ = 114 axons) vs. *Wnt1-Cre*$^{+/-}$,*Cbln1*$^{fl/fl}$ ($n$ = 191 axons), $^{****}p$ = 6.18 × 10$^{-37}$; *Wnt1-Cre*$^{+/-}$,*Cbln1*$^{fl/fl}$ + rhCbln1 ($n$ = 68 axons) vs. *Wnt1-Cre*$^{+/-}$,*Cbln1*$^{fl/fl}$ ($n$ = 191 axons), $^{****}p$ = 1.63 × 10$^{-21}$; *Cbln1*$^{fl/fl}$ vs. *Wnt1-Cre*$^{+/-}$,*Cbln1*$^{fl/fl}$ +rhCbln1, $^{**}p$ = 0.0098; by 1-way ANOVA followed by Tukey's multiple comparison test. Scale bar, 20 μm (D). (**F**) Neurogenesis is not disturbed in the *Cbln1* cKO cerebellum at P8. Immunostainings of the PC marker Calbindin and the granule cell marker NeuN showed no difference between *Cbln1* cKO and control cerebella, suggesting that the neurogenesis of PCs and GCs in cerebellum is not affected. Scale bars, 500 μm. (**G** and **H**) Lengths of PFs labeled by DiI were significantly decreased in *Cbln1* cKO mice at P6. The white arrowheads indicate the terminals of DiI-labeled PFs. Quantification of PF lengths is shown as box and whisker plots (H): $n$ = 190 axons for *Cbln1*$^{fl/fl}$ mice, $n$ = 132 axons for *Wnt1-Cre*$^{+/-}$,*Cbln1*$^{fl/fl}$ mice; $^{****}p$ = 1.09 × 10$^{-13}$; by unpaired Student $t$ test. Scale bar, 100 μm (G). The data underlying all the graphs shown in the figure are included in S1 Data. Cbln1, cerebellin 1; cKO, conditional knockout; GC, granule cell; h, hours; IGL, inner granule layer; ML, molecular layer; PC, Purkinje cell; PF, parallel fiber; rhCbln1, recombinant human Cbln1 protein.

organization of RGC axons in OC is critical for binocular vision [43]. We wanted to check whether Cbln1 contributed to axon guidance signaling in OC. RGCs do not express *Cbln1* (Fig 7A). However, both *Cbln1* mRNA and Cbln1 protein were detected in the ventral diencephalon at the floor of the third ventricle that is adjacent to OC (Fig 7B and 7C), implying that Cbln1 has the right location to exert effects on OC. In vitro co-culture of retinal explants with COS7 cell aggregates expressing Cbln1 showed that Cbln1 is sufficient to attract RGC axon turning (Fig 7D and 7E).

We next continued to explore whether Cbln1 physiologically regulates RGC axon guidance in OC at the ventral diencephalic midline. We generated *Cbln1* cKO embryos using *Nes-cre*. As show in Fig 7F, *Cbln1* expression in the ventral diencephalon was ablated. RGCs can be divided to ipsilateral and contralateral subgroups according to their projection laterality to the same or opposite side of the brain, respectively. The experiments checking expression of Cbln1 receptors in retina by in situ hybridization revealed that only *Nrxn1* and *Nrxn2* mRNA were detected in the developing retina (Fig 7G) while *Nrxn3*, *GluD1*, or *GluD2* mRNA was not detected (S7A and S7B Fig). Nrxn2 was further found to be only expressed in the contralateral RGCs marked by Brn3a (Fig 7H), suggesting that Cbln1 would only work on the contralateral RGCs. Consistent with this, DiI tracing of RGC axons showed that contralateral axon attraction to OC was impaired in the *Cbln1* cKO mouse embryos compared with control embryos (Fig 7I and 7J). We further checked the targeting of optic nerves to the brain by anterograde labeling with cholera toxin subunit B (CTB) and found that the ratio of ipsilateral area to contralateral area of the retinogeniculate projections was increased in *Cbln1* cKO pups compared with control pups (Fig 7K and 7L). These data suggest that the non-cell-autonomous Cbln1 regulates contralateral RGC axon guidance in the optic chiasm.

## Discussion

Based on the in vitro and in vivo studies in mice, we have demonstrated that cell-autonomous and non-cell-autonomous Cbln1 regulates axon growth and guidance in multiple neural regions, respectively, suggesting a general role for Cbln1 in early nervous system development.

### New roles of Cbln1 in axon development prior to synapse formation

Studies on Cbln1 so far have focused on its role as the synaptic organizer in cerebellum [19,20,32,37,38,44–46] and cortex [40]. Whether Cbln1 works in earlier neuronal developmental processes prior to synapse formation or in other neural regions is not known. Here, we report that Cbln1 is expressed in developing spinal cord, cerebellum, and ventral

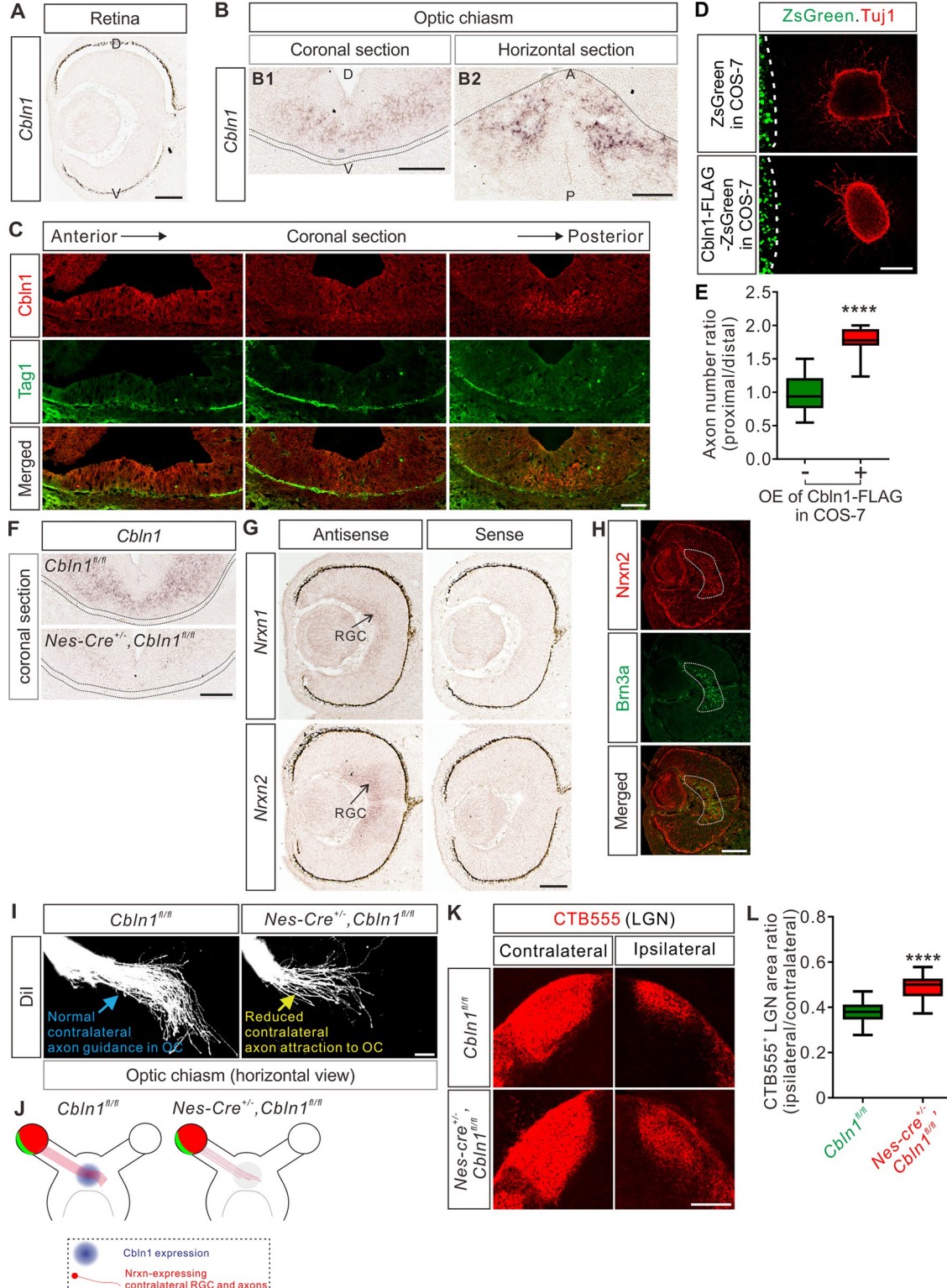

**Fig 7. Non-cell-autonomous Cbln1 regulates RGC (retinal ganglion cell) axon guidance in optic chiasm.** (**A**) In situ hybridization of *Cbln1* in the developing retina. *Cbln1* mRNA expression was not detected in the retina of E13 mouse embryos. D, dorsal; V, ventral. Scale bar, 100 μm.(**B**) In situ hybridization of *Cbln1* in E13 mouse brain sections (coronal section for B1; horizontal section for B2). *Cbln1* mRNA expression was detected in the floor of the third ventricle, adjacent to the optic chiasm. The dotted lines indicate the boundary of the optic chiasm. D, dorsal; V, ventral; A, anterior; P, posterior. Scale bars, 100 μm. (**C**) Immunostaining of

Cbln1 in coronal brain sections from E13 mouse embryos. Cbln1 protein was detected in the floor of the third ventricle, adjacent to the optic chiasm. Tag1-marked RGC axons projected to the optic chiasm. Serial sections from anterior level to posterior level were shown. Scale bar, 100 μm. (**D** and **E**) Extrinsic Cbln1 attracted RGC axon turning in vitro. Retina explants from E14.5 mouse embryos were co-cultured with COS7 cell aggregates expressing Cbln1-FLAG with ZsGreen or ZsGreen alone (D). Quantification of RGC axon turning was performed by measuring the axon number ratio (proximal/distal), similarly as CA guidance assay. Data are represented as box and whisker plots (E): Ctrl ($n$ = 13 explants) vs. OE ($n$ = 10 explants), $^{****}p$ = $4.59 \times 10^{-7}$; by unpaired Student $t$ test. Scale bar, 200 μm (D). (**F**) Ablation of *Cbln1* in *Nes-Cre$^{+/-}$,Cbln1$^{fl/fl}$* cKO mouse embryos was confirmed by in situ hybridization of E13 coronal brain sections. The dotted lines indicate the boundary of the optic chiasm. Scale bar, 100 μm. (**G**) In situ hybridization of *Nrxn1* and *Nrxn2* in E13 retina. *Nrxn1* and *Nrxn2* mRNAs were detected in retinal ganglion cells. Scale bar, 100 μm. (**H**) Immunostaining of Nrxn2 in E13 retina. Nrxn2 is expressed only in the contralateral RGCs marked by Brn3a. Scale bar, 100 μm. (**I** and **J**) Axon guidance defects in the OC of *Cbln1* cKO embryos. DiI tracing of RGC axons was performed to visualize axon trajectory in OC. Compared with normal axon attraction of contralateral RGCs in OC of control embryos, *Cbln1* cKO embryos showed reduced axon attraction to OC. The phenotype is illustrated in (J). Scale bar, 100 μm (I). (**K** and **L**) RGC central targeting defects of *Cbln1* cKO mice. Representative images of coronal sections through the LGN (lateral geniculate nucleus) after unilateral injection of CTB-Alexa Fluor 555 (CTB555) at P4 in *Cbln1* cKO and control mice were shown and projections to the contralateral and ipsilateral LGN are visible (K). Quantification of CTB555$^{+}$ "ipsilateral area"/"contralateral area" in LGN is represented as box and whisker plot (L): Ctrl ($n$ = 51 sections) vs. cKO ($n$ = 53 sections), $^{****}p$ = $2.06 \times 10^{-18}$; by unpaired Student $t$ test. Scale bar, 200 μm (K). The data underlying all the graphs shown in the figure are included in S1 Data. A, anterior; CA, commissural axons; Cbln1, cerebellin 1; D, dorsal; Ctrl, control; CTB555, CTB-Alexa Fluor 555; LGN, lateral geniculate nucleus; Nrxn, neurexin; OC, optic chiasm; OE, overexpression; P, posterior; RGC, retinal ganglion cell; V, ventral.

diencephalon. We also found that Cbln1 regulates axon growth and guidance in multiple neural regions. These findings suggest that Cbln1 has dynamic spatial–temporal expression patterns and functions in the nervous system. Thus, in order to distinguish the early (axon development) and late (synapse formation) roles of Cbln1, it would be critical to more precisely control the time point of knocking out *Cbln1*. For example, inducible *Cbln1* cKO would be necessary to explore its roles in synapse formation, in order to avoid disrupting its role in axon growth and pathfinding.

It is surprising that Cbln1-Neurexin 2 that has been shown to work as synaptic organizers is also required by the DCNs and other neurons for promoting growth and guidance when there are already known attractants and growth factors doing the same job. However, quite a few studies have shown that these synaptic organizers are also involved in earlier neural development including axon development. In *C. elegans*, Nrxn promotes neurite outgrowth of DVB neurons [47], and a neurexin-related protein, BAM-2, regulates axonal branches [48]. In *Drosophila*, homolog of α-neurexin (DNrx) restricts axonal branching [49], and neurexin and neuroligin-based adhesion molecules regulate axonal arborization growth independent of synaptic activity [50]. These studies suggest that these synapse organizers are also required in early neural development to regulate axon development, working together with those already known attractants and growth factors.

## Cbln1 has dual roles as both non-cell-autonomous and cell-autonomous cues to regulate axon guidance and growth, respectively

During neural developmental stages before synapse formation, extracellular cues are required to direct axon growth and guidance [51]. Most of these cues are non-cell-autonomous and secreted by sources such as the surrounding and target (intermediate or final) tissues. Here, we found that non-cell-autonomous Cbln1 expressed and secreted from FP in the developing spinal cord and from ventral diencephalon in the developing brain works in a paracrine manner to regulate CA and retinal ganglion axon guidance when they cross the midline.

In addition, we also found that cell-autonomous Cbln1 that is generated and secreted from commissural and cerebellar granule cell axons works in an autocrine manner to stimulate their own axon growth. Actually, other examples that axon-derived and remotely secreted cues regulate axon development have also been reported. The axonally secreted protein axonin-1

promotes neurite outgrowth of dorsal root ganglia (DRG) [52]. Wnt3a is expressed in RGCs and has the autocrine RGC axon growth-promoting activity [53]. The C terminus of the ER stress-induced transcription factor CREB3L2 was found to be secreted by DRG axons to promote DRG axon growth [54]. A recent study suggests that axonally synthesized Wnt5a is secreted and promote cerebellar granule axon growth in an autocrine manner [55].

Thus, Cbln1 shows up as an example of molecules with dual roles as both non-cell-autonomous and cell-autonomous cues to regulate axon guidance and growth, respectively.

## Cbln1-Nrxn signaling in axon development

We found that neurexins, especially Nrxn2, mediate Cbln1 functions in axon development. Neurexins are transmembrane proteins with a large extracellular region and a small intracellular C-terminal region [41]. *Nrxns* are alternatively spliced at 6 sites (named as SS1 to SS6) [56], whereas Cbln1 only bind to SS4+ neurexins [38]. Extracellularly, α-Nrxns bind to Cbln1 via the LNS6 domain (laminin/neurexin/sex-hormone-binding globulin domain 6) that is also shared by β-Nrxns [56]. Intracellularly, Nrxns interact with CASK, Mints, and protein 4.1 that nucleates actin cytoskeleton to regulate synapse formation [57–60]. It will be interesting to explore whether and how Cbln1-Nrxn signaling is mediated by intracellular CASK/Mint/ p4.1-cytoskeleton pathway to regulate axon growth and guidance in the early neuronal development.

## The cerebellin family

There are 4 members in the cerebellin family, Cbln1-4. Cbln1 and Cbln2 bind to Nrxn1-3 and GluD1-2, while Cbln4 is a highly specific ligand for DCC [61]. In addition, *Cbln4*-null mice did not show any defect in commissural axon guidance in the developing spinal cord [61].

Cbln3 has only been studied in synapse structural and functional regulation and is secreted only when it is bound to Cbln1 [62]. Cbln1 and Cbln3 proteins are both lost in *Cbln1*-null mice, while *Cbln3*-null mice have a robust increase in Cbln1 protein level [62]. It would be interesting to check if Cbln3 is expressed in the developing dorsal spinal cord and brain, and if there is a similar Cbln1-Cbln3 interaction mechanism in the earlier embryonic stages.

It has been reported that Cbln1 and Cbln2 have redundancy in cerebellum but not in thalamic neurons in the adult mice [63], supporting a tissue-specific redundancy in adulthood. Cbln2 is expressed in the developing chick dorsal spinal cord [64]. *Cbln1-Cbln2* double cKO may be useful to check their redundancy in the developing brain regions and axon developmental processes described in our current study.

## Materials and methods

### Ethics statement

All experiments using mice were carried out following animal protocols (SUSTC-JY2017004) approved by the Laboratory Animal Welfare and Ethics Committee of Southern University of Science and Technology.

### Animals

Generation of *Cbln1* cKO mice was performed following procedures described previously [65], with the whole coding sequence as the targeted region (S2A Fig). *Cbln1*$^{+/fl}$ mice and corresponding *Cre* mice lines were used to generate *Cbln1* cKO and littermate control embryos. Genotyping primers are as following: the first *Cbln1-loxP* site, 5′-ACGCGGGGACATTTGT TCTGGAGT-3′ and 5′-ACGATGGGCTCTGTCTCATTCTGC-3′; the second *Cbln1-loxP* site,

5′-AGAAAGGCGACCGAGCATAC-3′ and 5′-AGTGTGCAGAGCTAAGCGAA-3′. *Wnt1-cre* [21], *Rosa26mTmG* [23], *Gli2$^{+/-}$* [26], *Foxa2-cre$^{ERT}$* [36], and *Nes-cre* [66] mice used in this study were described in the indicated references, and their stock numbers in The Jackson Laboratory are 003829, 007676, 007922, 008464, and 003771, respectively. All mice were housed in a specific pathogen-free animal facility at the Laboratory Animal Center of Southern University of Science and Technology. For timed pregnancy, embryos were identified as E0.5 when a copulatory plug was observed at noon. To induce Cre activity for *Foxa2-cre$^{ERT}$*-derived *Cbln1* cKO in FP, 8-mg tamoxifen (Cayman Chemical) was given orally to E8.5 pregnant mice with an animal gauge feeding needle. Fertilized chick eggs were purchased from a local supplier and chick embryos developed in an incubator (BSS 420, Grumbach) were staged using the Hamburger and Hamilton staging system. For all experiments with mice or chick, a minimum of 3 (up to 20) embryos or pups was analyzed for each genotype or experimental condition.

### In ovo electroporation

The chick spinal DCN-specific knockdown vector pMath1-eGFP-miRNA was a gift from Esther T. Stoeckli [28]. DCN-specific overexpression vector pMath1-eGFP-IRES-MCS was constructed by replacing the miRNA cassette with an IRES sequence plus multiple coning sites (MCS). The coding sequence of chick *Cbln1* was cloned from St. 23/24 chick spinal cord cDNA with primers 5′-ATGCGGGGCCCG-3′ and 5′-TTAAAGCGGGAACACC-3′. In ovo electroporation was carried out using the ECM 830 Square Wave Electroporator (BTX) as previously described [67]. Electroporation was performed at St.17 and embryos were collected and analyzed at St.23.

### Tissue and neuron culture

All tissue culture reagents were from Thermo unless otherwise specified. DCN explant and neuronal culture were carried out as describe previously [65]. The working concentration for recombinant human Cbln1 (R&D Systems) was 500 ng/ml. GPN (Abcam) was dissolved in DMSO and used at the working concentration of 50 μm. Bafilomycin A1 (Coolaber) was dissolved in DMSO and used at the working concentration of 200 nM. P6-P8 mouse cerebella were cut into small pieces with scissors after the meninges were carefully removed. The tissue was then digested in 5 ml HBSS containing 0.1% Trypsin and 0.04% DNase I in a 37°C water bath for 15 min before termination addition of 5 ml BME with 10% FBS. Cell suspension was obtained by filtering with sterile cell strainers (40 μm). After centrifuged at 200×g for 5 min, the cell pellets were resuspended in BME supplemented with 5% FBS, 1× GlutaMAX-1, 0.5% glucose, and 1× penicillin/streptomycin. The neurons were then plated in PDL-coated cell culture plate and the medium was replaced by maintenance medium supplemented with 1× B27, 1× GlutaMAX-1, 0.5% glucose, and 1× penicillin/streptomycin after 4 h.

### Knockdown or overexpression using lentiviral system, RT-qPCR, and western blotting

The lentiviral knockdown constructs were made using pLKO.1-Puro or pLKO.1-GFP plasmids (Addgene) [65]. The target sequences of shRNA are as following: sh*Cbln1*, 5′-GGCTGGAAG TACTCAACCTTC-3′; sh*Nrxn2*, 5′-CGTTCGTTTATTTCCCTCGAT-3′; sh*Nrxn2α*, 5′-GGA CTTCTGCTGTTCAACTCA-3′; sh*Nrxn2β*, 5′-CTCCCCCATCACCCGGATTTG-3′; *shCtrl for* pLKO.1-Puro system: 5′-GCATCAAGGTGAACTTCAAGA-3′; *shCtrl for* pLKO.1-GFP system: 5′-GCATAAACCCGCCACTCATCT-3′. RT-qPCR was performed as previously reported [65]. Primers used in qPCR are as following: *mCbln1*, 5′-CCGAGATGAGTAATCG

CACCA-3′ and 5′-TCAACATGAGGCTCACCTGGATG-3′; *mNrxn2*, 5′-TACCCGGCAGGA AACTTTGA-3′ and 5′-CCCCCTATCTTGATGGCAGC-3′; *mNrxn2α*, 5′-CTCAAGTCTGGG GCTGTCTG-3′ and 5′-ATAGCGTGTCCAATCCCTGC-3′; *mNrxn2β*, 5′-GATGGATCCAG GCTTCACGG-3′ and 5′-GAAGGAAAACCAGAGCCCGA-3′; *mGapdh*: 5′-TTGTCAGCAA TGCATCCTGCACCACC-3′ and 5′-CTGAGTGGCAGTGATGGCATGGAC-3′.

The coding sequence of mouse *Cbln1* was cloned from E11.5 mouse spinal cord cDNA with primers 5′-CCGGAGGCGCGATGCT-3′ and 5′-ATTCCCGATACGTGCCAG-3′, and lenti viral expression construct was constructed using the pHBLV-CMV-MCS-3×Flag-EF1-Zsgreen1-T2A-Puro backbone (Hanbio). After infection with the lenti virus, the COS7 cell line stably expressing Cbln1 was acquired after multiple rounds of selection using puromycin. Expression of Cbln1-FLAG in cell pellets and supernatant was validated by western blotting (WB) following the standard protocols. The dilutions and sources of antibodies used in WB are as following: Cbln1 (1:100, Abclonal), FLAG (1:1,000, Beyotime), and β-actin (1:10,000, Abcam).

## Identification of the differentially expressed genes in the dorsal spinal cord of mouse embryos

*Wnt1-cre* and *Rosa26mTmG* mice were mated to generate *Wnt1-Cre,Rosa26mTmG* embryos. E10.5, E11.5, and E12.5 embryos were collected and dissected, and dorsal spinal cords were dissociated and GFP$^+$ neurons were purified using FACS. RNAs were prepared from these purified neurons and the expression profiling was carried out using microarray analysis with GeneChip Mouse Exon 1.0 ST Array (Affymetrix) following the manufacturer's manual.

## In situ hybridization

In situ hybridization using DIG-labeled RNA probes was carried out on sections from mouse or chick tissue sections following a previously reported protocol [68]. The primers used for PCR in cloning the templates for generating RNA probes are as following (all mouse clones except indicated): *Cbln1*, 5′-CCAAGACGTGACACGCGAGG-3′ and 5′-CAGTAAGTGGCA GGGTTCAG-3′; chick *Cbln1*, 5′-GAGAAGACGCCGCTCAGGTGT-3′ and 5′-CGGGTTGA TTTGCGGTCCTTC-3′; *Nrxn1*, 5′-CAGGGAATGCGATCAGGAGG-3′ and 5′-AGACTTCT TCTCTGGCACGC-3′; *Nrxn2*, 5′-TCACAGCCCTGGGTTGATTT-3′ and 5′-AGCAGCGAC ACACACAAAAG-3′; *Nrxn3*, 5′-GTGAGATGGGGTGTACCACG-3′ and 5′-ACACACACA CTGGTCAGAACC-3′; *GluD1*, 5′-CATTGGCCTCCTTCTTGCCT-3′ and 5′-GAGGTGCCA TGAGAGGTGTC-3′; *GluD2*, 5′-GCCCCTACCGTGATGTCTTT-3′ and 5′-GTCAATGTC CAGAGGGGTCA-3′; *Nrxn2α*, 5′-GCAGGGATTGGACACGCTAT-3′ and 5′-GAACTGT GACTGCCTACCCC-3′. The template for *Nrxn2β*-specific RNA probe was synthesized (5′-TGAGGGGGGACCCCTAGCCGCCCGCGATGGATCCAGGCTTCA

CGGACCTTGGCCTTCCCGCTGCGCGTACCCCGGATTCCCCGGCGGGATCCAGT TGATTTGCTTGGCTCCGGACTGAGGCTCGGGCTCTGGTTTTTCCTTCGCTTCACCCC TACCCCCCTCTCGGAGCTCGCAACCGGAGGGGGGCTTT-3′) and cloned to pUC57 (San-gon). RNA probes were transcribed in vitro using DIG RNA Labeling Kit (SP6/T7) (Roche). Anti-Digoxigenin-AP and NBT/BCIP Stock Solution were also from Roche. In situ hybridization images were collected with Axio Imager A2 (Zeiss) or TissueFAXS Cytometer (TissueGnostics).

## Axon guidance assay using co-culture of COS7 cell aggregates with DCN or retinal explants

Aggregates of COS7 cells stably expressing Cbln1 were prepared by resuspending cells in rat tail collagen gel and then placed into 24-well glass bottom plates (Nest). DCN explants were

dissected from E11 mouse embryos, immersed in collagen gel, and placed 200 to 400 µm away from the COS7 aggregates. Explants and cell aggregates were co-cultured for 40 to 48 h in neurobasal medium supplemented with B27, GlutaMAX-1 and penicillin/streptomycin. Similarly, retinal explants were dissected from E14.5 mouse retinas and co-cultured with COS7 aggregates for around 30 h, with the culture medium as following: Neurobasal medium mixed with DMEM/F12 (1:1), supplemented with B27, N21 MAX Media Supplement, GlutaMAX-1 and penicillin/streptomycin.

## DiI tracing of axons

*FAST* DiI (Invitrogen, D7756) tracing of commissural axons in the spinal cord was performed as previously reported [65]. *FAST* DiI (Invitrogen, D7756) labeling of cerebellar parallel fibers was performed as previously described [69]. DiI (Invitrogen, D282) tracing of optic nerves was performed as previously reported [70].

## CTB labelling of optic nerve

To label RGC axon terminals in P4 mouse brain, RGC axons were anterogradely labeled by CTB (Cholera Toxin Subunit B) conjugated with Alexa Fluor 555 (Invitrogen, C34776) through intravitreal injection 24 h before sacrifice. After PFA perfusion, the brains were fixed with 4% PFA in 0.1 M PB overnight, dehydrated with 15% sucrose and 30% sucrose in 0.1 M PB overnight at 4°C sequentially, embedded with O.C.T. for coronal section, and cryosectioned at 12 µm with Leica CM1950 Cryostat. The images were captured on Tissue Genostics with identical settings for each group in the same experiment with the TissueFAXS 7.0 software.

## Immunostaining and immunofluorescence

Immunostaining of tissue sections and IF of cultured DCN explants and neurons were done as previously described [65]. The spinal cord open-books were prepared similarly as DiI tracing, and their immunostaining was performed with similar procedures as tissues sections except that all incubation and washing were done in 24-well plates and the open-books were mounted onto slides and covered with cover slips before confocal imaging. For IF of axon surface Cbln1 in cultured DCN neurons after GPN, Bafilomycin A1 or sh*Nrxn2* treatment, the permeabilization step was omitted and there was no Triton x-100 in antibody incubation buffers. Immunostaining of tissue sections from the cerebellum and the optic chiasm were done as previously described [44,70]. The dilutions and sources of antibodies used in immunostaining and immunofluorescence are as following: Cbln1 (1:1,000, Abcam), Cbln1 (1:50, Frontier Institute), Lhx2 (1:500, Abcam), Alcam (1:200, R&D Systems), Lhx9 (1:50, Santa Cruz Biotechnology), Robo3 (1:500, R&D Systems), Tag1 (1:200, R&D Systems), NFM (1:1,000, Cell Signaling Technology), GFP (1:1,000, Abcam), Isl1/2 (1:500, DSHB), Nrxn2 (1:200, Abcam), Calbindin (1:200, Swant), NeuN (1:500, Cell Signaling Technology), Brn3a (1:200, Millipore), Lim1/2 (1:10, DSHB). Alexa Fluor-conjugated secondary antibodies (Thermo) were used at 1:1,000 (555) or 1:500 (488). Fluorescent images were acquired using laser-scanning confocal microscopes Nikon A1R with NIS software, Leica SP8 with LASX software, or Zeiss LSM 800 with Zen software. All images were collected with identical settings for each group in the same experiment. Quantification of immunofluorescence signals was performed using ImageJ.

## Statistical analysis

Statistical analysis was performed with GraphPad Prism 7.0. Most of our data are represented as the box and whisker plots unless otherwise specified in the figure legends, and the settings

are: 25 to 75 percentiles (boxes), minimum and maximum (whiskers), and medians (horizontal lines). Unpaired Student $t$ test was performed for comparison between 2 groups and ANOVA with Tukey's multiple comparison test was performed to the comparison of 3 or more groups. $*$ Indicated statistically significant: $^*p < 0.05$, $^{**}p < 0.01$, $^{***}p < 0.001$, $^{****}p < 0.0001$.

## Supporting information

**S1 Fig. Screening of the differentially expressed genes in the mouse embryonic dorsal spinal cord. (Related to Fig 1).** (**A**) The embryonic dorsal spinal neurons were genetically labeled with eGFP by crossing *Wnt1-cre* with *Rosa26mTmG* mice. Immunofluorescence of cross-sections of E11.5 spinal cord was shown. DCN, dorsal commissural neurons. CA, commissural axons. Scale bar, 100 μm. (**B**) The dissected E11.5 spinal cords were shown in both bright-field and fluorescent images. The regions in the red dotted boxes were shown with higher magnification in the lower images. The dotted white line indicates where to cut and separate dorsal and ventral spinal cord. (**C**) The scheme showing the procedures for identifying the differentially expressed genes in the mouse embryonic dorsal spinal cord. (**D** and **E**) Co-immunostaining of Cbln1 with Lim1/2 (D) or Brn3a and Isl1/2 (E) in spinal cord cross-sections at E11 showed expression of Cbln1 in the DCNs. Circled areas highlight the expression of Cbln1 in the dI2 neurons marked by Lim1/2, the dI3 neurons co-labeled by Brn3a and Isl1/2, and the Brn3a$^+$ dI4 neurons just below dI3. Scale bars, 50 μm. (**F**) Co-immunostaining of Cbln1 with Alcam in spinal cord cross-sections of *Gli2* KO and its littermate control embryos at E11.5. The circled areas indicate the expression and loss of Cbln1 in control and *Gli2* KO embryos, respectively. Scale bar, 50 μm. DCN, dorsal commissural neurons; CA, commissural axon; FACS, Fluorescence-activated cell sorting; DEG, differentially expressed genes; dI2, dI3, dI4, dorsal interneuron 2, 3, 4.
(TIF)

**S2 Fig. DCN-specific cKO of *Cbln1* does not affect neurogenesis and patterning of spinal DCNs. (Related to Fig 2).** (**A**) Lhx2 and Lhx9 immunostaining in E11.5 spinal cord indicated that *Cbln1* cKO in DCNs does not disturb neurogenesis or patterning of the dI1 neurons. Scale bar, 100 μm. (**B** and **C**) Quantification of Lhx2$^+$ and Lhx9$^+$ neurons in (A) showed that the dI1 neurogenesis is not affected in *Cbln1* cKO. All data are represented as box and whisker plots: *Cbln1$^{fl/fl}$* ($n = 20$ sections) vs. *Wnt1-Cre$^{+/-}$,Cbln1$^{fl/fl}$* ($n = 16$ sections); ns, not significant ($p = 0.46$ for Lhx2$^+$ neurons in B, $p = 0.99$ for Lhx9$^+$ neurons in C); by unpaired Student $t$ test. (**D** and **E**) Lim1/2 and Brn3a/Isl1/2 immunostaining in E11.5 spinal cord indicated that *Cbln1* cKO in DCNs does not affect neurogenesis or patterning of the dI2, dI3, or dI4 neurons. Scale bars, 50 μm. (**F–H**) Quantification of dl2, dl3, and dl4 neurons in (D and E) showed that their neurogenesis is not affected. All data are represented as box and whisker plots: *Cbln1$^{fl/fl}$* ($n = 36$ sections for F, $n = 33$ sections for G and H) vs. *Wnt1-Cre$^{+/-}$,Cbln1$^{fl/fl}$* ($n = 38$ sections for F–H); ns, not significant ($p = 0.37$ for dl2 neurons in F, $p = 0.31$ for dl3 neurons in G, $p = 0.79$ for dl4 neurons in H); by unpaired Student $t$ test. (**I**) Commissural axon growth could catch up at later stages in DCN-specific *Cbln1* cKO in vivo. Commissural axons were marked by Robo3 immunostaining in spinal cord open-books at E12, showing that almost all Robo3-labeled commissural axons in *Cbln1* cKO reached the floor plate. Scale bar, 500 μm. (**J** and **K**) Quantification of commissural axon lengths and numbers in (I). The spinal cords were divided to bins (500 μm) along the anterior-posterior (A➔P) direction. All data are mean ± SEM: *Cbln1$^{fl/fl}$* ($n = 8$ embryos) vs. *Wnt1-Cre$^{+/-}$,Cbln1$^{fl/fl}$* ($n = 10$ embryos). All bins show no difference in axon length (J) or number (K) by multiple $t$ tests. (**L**) DiI labeling of spinal cord open-books at E12 showed that crossing or post-crossing commissural axons are behaving normally

in DCN-specific *Cbln1* cKO embryos compared with controls. Scale bar, 50 μm. The data underlying all the graphs shown in the figure are included in S1 Data. Cbln1, cerebellin 1; DCNs, dorsal commissural neurons; cKO, conditional knockout; dI1, dI2, dI3, dI4, dorsal interneuron 1, 2, 3, 4; A➜P, anterior to posterior.
(TIF)

**S3 Fig. Knockdown of mCbln1 using lentiviral shRNA and inhibition of axonal Cbln1 secretion by BafA$_1$ treatment. (Related to Fig 3). (A)** Validation of knockdown of *Cbln1* using lentiviral sh*Cbln1*. Since Cbln1 is abundant in cerebellar granule cells, we prepared dissociated cerebellar granule cells from P8 mouse pups and cultured in vitro to test the knockdown efficiency of sh*Cbln1*. *Cbln1* mRNA levels were measured by RT-qPCR after lentiviral shRNA infection. Data are mean ± SEM and represented as dot plots: $^{**}p = 0.0070$; by unpaired Student *t* test. **(B)** Bafilomycin A1 (BafA$_1$) blocked lysosomal exocytosis of Cbln1 from commissural axons. Cbln1 IF signals were reduced on the commissural axon surface of cultured DCN explants after 200 nM BafA$_1$ treatment for 4 h. Scale bar, 50 μm. **(C)** Quantification of axon surface Cbln1 IF signals in (B). Data are represented as box and whisker plots: Vehicle ($n = 182$ axons) vs. BafA$_1$ ($n = 220$ axons), $^{****}p = 5.8 \times 10^{-106}$, by unpaired Student *t* test. The data underlying all the graphs shown in the figure are included in S1 Data. Cbln1, cerebellin 1; BafA$_1$, Bafilomycin A1; shCtrl, control shRNA; sh*Cbln1*, shRNA against *Cbln1*; SEM, standard error of the mean; DCNs, dorsal commissural neurons; IF, immunofluorescence; A. U., arbitrary unit.
(TIF)

**S4 Fig. Floor plate-specific *Cbln1* cKO does not disturb neural patterning or neurogenesis in the developing spinal cord. (Related to Fig 4). (A)** Overexpression and secretion of Cbln1 tagged by FLAG in COS7 cells were validated by WB. **(B and C)** Isl1/2 immunostaining showed normal patterning of spinal cord in the floor plate-specific *Cbln1* cKO embryos (B). Isl1/2 marks different interneurons and motor neurons in spinal cord. The data for quantification of Isl1/2$^+$ neuron numbers are represented as box and whisker plots (C): *Cbln1$^{fl/fl}$* ($n = 42$ sections) vs. *Foxa2-Cre$^{+/-}$,Cbln1$^{fl/fl}$* ($n = 49$ sections); ns, not significant ($p = 0.93$); by unpaired Student *t* test. Scale bar, 100 μm (B). **(D and E)** The floor plate-specific cKO of Cbln1 does not disturb DCN neurogenesis, spinal cord patterning, or floor plate development. Lhx2 and Alcam immunostainings of E11.5 spinal cord were used to mark dI1 commissural neurons and floor plate, respectively (D). The data for quantification of Lhx2$^+$ neuron numbers are represented as box and whisker plots (E): *Cbln1$^{fl/fl}$* ($n = 16$ sections) vs. *Foxa2-Cre$^{+/-}$,Cbln1$^{fl/fl}$* ($n = 35$ sections); ns, not significant ($p = 0.59$); by unpaired Student *t* test. Scale bar, 100 μm (D). **(F)** The axon guidance defects of pre-crossing commissural axons were observed by NFM immunostaining in floor plate-specific *Cbln1* cKO and control embryos at E11.5. Higher magnification views of the FP region in the white dotted boxes are also shown (bottom). The pair of red arrowheads denotes the thickness of the VC. The double-arrowed line measures the distance between the point of intersection (of the main pre-crossing commissural axon bundle with the ventral edge of spinal cord) and the midline (indicated by the dotted line). Scale bars, 50 μm. **(G and H)** Quantification of the VC thickness and the distance from the main bundle intersection point to the midline. All data are represented as box and whisker plots: *Cbln1$^{fl/fl}$* ($n = 49$ sections) vs. *Foxa2-Cre$^{+/-}$,Cbln1$^{fl/fl}$* ($n = 74$ sections), $^{****}p = 3.86 \times 10^{-5}$ for G, $^{****}p = 1.35 \times 10^{-7}$ for H, by unpaired Student *t* test. The data underlying all the graphs shown in the figure are included in S1 Data. Cbln1, cerebellin 1; DCNs, dorsal commissural neurons; Ctrl, control; OE, overexpression; cKO, conditional knockout; dI1, dorsal interneuron 1; NFM, neurofilament; FP, floor plate; VC, ventral commissure.
(TIF)

**S5 Fig. Working models for cell-autonomous and non-cell-autonomous Cbln1 in the developing spinal cord. (Related to Fig 5).** (**A**) *GluD1 or GluD2* mRNA was not detected in E11.5 spinal cord cross-sections by in situ hybridization. Scale bar, 100 μm. (B–D) Validation of knockdown by shRNAs. Dissociated cerebellar granule cells from P8 mouse pups were cultured and lentiviral shRNAs were infected. Significant knockdown was achieved by shRNAs against *Nrxn2*, *Nrxn2α*, and *Nrxn2β*, respectively. RT-qPCR data are mean ± SEM and represented as dot plots: $^{**}p = 0.0084$ for B; $^{***}p = 0.00010$ for C; $^{***}p = 0.00020$ for D; by unpaired Student $t$ test. (E) Surface Cbln1 IF signals were reduced in the commissural axons and growth cones after lentiviral sh*Nrxn2* infection. Dissociated DCN neurons from E11 mouse embryos were cultured and infected by lentiviral sh*Nrxn2*. Surface Cbln1 IF signals were imaged. Scale bar, 5μm. (F) Quantification of axon surface Cbln1 IF signals in (E). Data are represented as box and whisker plots: shCtrl ($n = 27$ axons) vs. sh*Nrxn2* ($n = 26$ axons), $^{****}p = 1.13 \times 10^{-12}$, by unpaired Student $t$ test. (G) Working model for the stimulation of commissural axon growth by the cell-autonomous Cbln1. In the pre-crossing commissural axons, Cbln1 is expressed cell-autonomously by the dorsal commissural neurons and axons. Commissural axon growth cone-secreted Cbln1 works back to itself in an autocrine manner and binds to Nrxn2 receptors to stimulate commissural axon growth. In the DCN-specific *Cbln1* cKO embryos, commissural axon growth is reduced compared with their littermate controls. (H) Working model for the attraction of commissural axon growth toward midline by the non-cell-autonomous, floor plate-derived Cbln1. When commissural axons approach the midline, the floor plate-derived Cbln1 attracts commissural axons to the midline that is also mediate by Nrxn2 receptors. In the floor plate-specific *Cbln1* cKO embryos, commissural axon guidance in the midline crossing is impaired, resulting in a U-shaped and thinner ventral commissure compared with the V-shaped and thick ventral commissures in the littermate control embryos. The data underlying all the graphs shown in the figure are included in S1 Data. Cbln1, cerebellin 1; GluD1, GluD2, glutamate receptor delta 1, 2; Nrxn, neurexin; SEM, standard error of the mean; DCNs, dorsal commissural neurons; IF, immunofluorescence; shCtrl, control shRNA; sh*Nrxn*, shRNA against *neurexin*.
(TIF)

**S6 Fig. Expression of Cbln1 receptors in the developing cerebellum. (Related to Fig 6).** (**A** and **B**) In situ hybridization of *Nrxn1*, *Nrxn2*, *Nrxn3*, *GluD1*, and *GluD2* in cerebella at P8 (A) and P15 (B). *Nrxn1*, *Nrxn2*, and *Nrxn3* mRNAs were detected in the IGL. *GluD2* mRNA was highly and specifically expressed in the PCs while *GluD1* mRNA was not detected in the cerebellum at these stages. Scale bars, 500 μm. Cbln1, cerebellin 1; Nrxn, neurexin; GluD1, GluD2, glutamate receptor delta 1, 2; IGL, inner granule layer; PC, Purkinje cells.
(TIF)

**S7 Fig. In situ hybridization of Cbln1 receptors in the developing retina. (Related to Fig 7).** (**A** and **B**) In situ hybridization of *Nrxn3* (A), and *GluD1* and *GluD2* (B) in E13 retina. *Nrxn3*, *GluD1*, or *GluD2* mRNA was not detected in the developing retina. Scale bars, 100 μm. Cbln1, cerebellin 1; Nrxn3, neurexin 3; GluD1, GluD2, glutamate receptor delta 1, 2.
(TIF)

**S1 Table. Differentially expressed genes in the dorsal spinal cord of mouse embryos. (Related to Fig 1).**
(XLS)

**S1 Raw Images. Full western blot images for S4A Fig.**
(PDF)

**S1 Data. Raw data and statistical data analysis for all the graphs.**
(XLSX)

## Acknowledgments

We thank Esther T. Stoeckli for the pMath1-eGFP-miRNA vector, members of Ji and Jaffrey laboratories for help, technical support, and comments on the manuscript. We thank the technical support from the Laboratory Animal Center and the Core Research Facilities of Southern University of Science and Technology.

## Author Contributions

**Conceptualization:** Peng Han, Samie R. Jaffrey, Sheng-Jian Ji.

**Data curation:** Peng Han, Yuanchu She, Samie R. Jaffrey, Sheng-Jian Ji.

**Formal analysis:** Peng Han, Yuanchu She, Sheng-Jian Ji.

**Funding acquisition:** Samie R. Jaffrey, Sheng-Jian Ji.

**Investigation:** Peng Han, Yuanchu She, Zhuoxuan Yang, Mengru Zhuang, Qingjun Wang, Xiaopeng Luo, Chaoqun Yin, Junda Zhu, Sheng-Jian Ji.

**Methodology:** Peng Han, Yuanchu She, Zhuoxuan Yang, Mengru Zhuang, Xiaopeng Luo, Chaoqun Yin, Junda Zhu, Samie R. Jaffrey, Sheng-Jian Ji.

**Project administration:** Samie R. Jaffrey, Sheng-Jian Ji.

**Resources:** Samie R. Jaffrey, Sheng-Jian Ji.

**Supervision:** Samie R. Jaffrey, Sheng-Jian Ji.

**Validation:** Peng Han, Yuanchu She, Zhuoxuan Yang, Sheng-Jian Ji.

**Visualization:** Peng Han, Yuanchu She, Sheng-Jian Ji.

**Writing – original draft:** Peng Han, Yuanchu She, Sheng-Jian Ji.

**Writing – review & editing:** Peng Han, Yuanchu She, Samie R. Jaffrey, Sheng-Jian Ji.

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
