## [Editor Report · Decision Letter 0]

21 Apr 2022

Dear Dr Ji, 

Thank you for submitting your manuscript entitled "Cbln1 regulates axon growth and guidance in multiple neural regions" for consideration as a Research Article by PLOS Biology. Please accept my apologies for the delay in providing you with our decision.

Your manuscript has now been evaluated by the PLOS Biology editorial staff as well as by an academic editor with relevant expertise and I am writing to let you know that we would like to send your submission out for external peer review.

Once your full submission is complete, your paper will undergo a series of checks in preparation for peer review. Once your manuscript has passed the checks it will be sent out for review. To provide the metadata for your submission, please Login to Editorial Manager (https://www.editorialmanager.com/pbiology) within two working days, i.e. by Apr 25 2022 11:59PM.

If your manuscript has been previously reviewed at another journal, PLOS Biology is willing to work with those reviews in order to avoid re-starting the process. Submission of the previous reviews is entirely optional and our ability to use them effectively will depend on the willingness of the previous journal to confirm the content of the reports and share the reviewer identities. Please note that we reserve the right to invite additional reviewers if we consider that additional/independent reviewers are needed, although we aim to avoid this as far as possible. In our experience, working with previous reviews does save time. 

If you would like to send previous reviewer reports to us, please email me at ialvarez-garcia@plos.org to let me know, including the name of the previous journal and the manuscript ID the study was given, as well as attaching a point-by-point response to reviewers that details how you have or plan to address the reviewers' concerns. 

Kind regards,

Ines

--

Ines Alvarez-Garcia, PhD

Senior Editor

PLOS Biology

---

## [Decision Letter · Decision Letter 1]

16 Jun 2022

Dear Dr Ji,

Thank you for your patience while your manuscript entitled "Cbln1 regulates axon growth and guidance in multiple neural regions" was peer-reviewed at PLOS Biology and please accept my apologies for the delay in providing you with our decision. It has now been evaluated by the PLOS Biology editors, an Academic Editor with relevant expertise, and by two independent reviewers. 

As you will see, the reviewers find your conclusions novel and interesting, although they also think that more work needs to be done to confirm the findings and would like you to add more details on the mechanisms underlying Cbln1 role in axon guidance. After consulting with the rest of the team and the Academic Editor, we agree with the interest of the findings and would like to invite you to submt a revision that addresses the outstanding issues.

We cannot make a decision about publication until we have seen the revised manuscript and your response to the reviewers' comments. Your revised manuscript is likely to be sent for further evaluation by all or a subset of the reviewers.

**IMPORTANT - SUBMITTING YOUR REVISION**

3. Resubmission Checklist

a) *PLOS Data Policy*

b) *Published Peer Review*

d) *Blurb*

Please also provide a blurb which (if accepted) will be included in our weekly and monthly Electronic Table of Contents, sent out to readers of PLOS Biology, and may be used to promote your article in social media. The blurb should be about 30-40 words long and is subject to editorial changes. It should, without exaggeration, entice people to read your manuscript. It should not be redundant with the title and should not contain acronyms or abbreviations. For examples, view our author guidelines: https://journals.plos.org/plosbiology/s/revising-your-manuscript#loc-blurb

Sincerely,

Ines

--

Ines Alvarez-Garcia, PhD

Senior Editor

PLOS Biology

Reviewers' comments

Rev. 1:

The authors of this study investigated a novel role for the protein Cerebellin-1 (Cbln1) in axon guidance during development. Using microarray analysis of dorsal spinal cord neurons, Cbln1 was identified as one of the differentially expressed gene candidates and verified by in situ hybridization that its mRNA is expressed both in dorsal spinal commissural neurons and the ventral floor plate. In order to study the cell autonomous (dorsal commissural neurons) and non-cell autonomous (floor plate) roles of Cbln1, a conditional flox Cbln1 mouse was generated and crossed with tissue-specific Cre lines. Using in vivo mouse genetics for loss-of-function, in utero electroporation in the chick for gain-of-function and in vitro tissue culture assays, it was shown that Cbln1 promoted commissural axon growth cell autonomously in commissural neurons and guidance by the floor plate in a non-cell autonomous manner. The investigators go on to show that Cbln1 most likely mediates these axon guidance events through it's receptor Neurexin-2. It is also interesting that Cbln1 signaling mediates axonal growth and guidance in other neuronal populations including cerebellar granule cells and retinal ganglionic cells, of which these results suggest that Cbln1 is used widely by the nervous system for axon growth and guidance besides its previously known role in synaptogenesis. This study adds new knowledge to the complex and diverse molecular mechanisms involved in axon guidance during early nervous system development. However, this study falls short of demonstrating the downstream signaling mechanism employed by Cbln1 in axon guidance. It is also not known whether the same or different signaling mechanisms are used by Cbln1 in synaptogenesis versus axon guidance, which would add substantial significance to this work.

Here are a few issues that the authors should address:

1. Higher magnifications images should also be presented of the open-book preparation for the WT and conditional mutant in Fig. 2B to better illustrate what the authors claimed that there are fewer number of commissural axons and shorter axons, especially when it is not clear in the quantification shown in Fig. 2D whether individual axons or bundled of axons are actually counted.

2. Fig. 3F and 3H show the in vitro commissural axon cultures treated with GPN to block Cbln1 exocytosis resulting in slower axon growth rate. First, the image panels in 3F does not appear to be different in terms of illustrating axon growth rate. Second, did the investigators look at whether the lack of Cbln1 signaling in the axons, especially in the in vivo conditional knockout mutants, only slow down the growth rate and that the mutant axons will eventually catch-up. This would make sense as there are other attractant and growth promoting molecules used by the dorsal spinal commissural neurons, like Netrin-DCC.

3. While this reviewer appreciates the brevity of the Discussion, in this case it might be a bit lacking. There is no discussion or speculation as to why Cbln1-Neurexin 2 is required by the dorsal commissural axons or other neurons and their axons for promoting growth and guidance when there are already known attractants and growth factors doing the same job. Is Cbln1-Neurexin 2 required for a specific subset of commissural axons as there are at least 4 different types, dI1-dI4, in the dorsal rodent spinal cord during development.

4. Finally, a minor correction is needed on Page 13 in the subsection "Cbln1-Nrxn signaling in axon development", first sentence: "esp." needs to be spelled out.

Rev. 2:

This manuscript by Han et al., identifies Cerebellin-1 as an attractive guidance cue for axons from several populations of neurons, including dorsal commissural neurons, retinal ganglion cells, and cerebellar granule neurons. Identification of a novel guidance cue is significant to the field. Previously, Cerebellin-1 had only been examined in synapse formation. Here, the authors use both mouse and chick models to demonstrate a role for Cerebellin-1 in axon guidance. They find evidence for both cell autonomous functions of the protein (promoting outgrowth in dorsal commissural neurons), as well as non-cell autonomous roles of Cerebellin-1 secretion from the floor plate and the ventral diencephalon to guide axons. Overall, the manuscript convincingly defines such a broad and conserved role. I have one suggested experiment, and several points for clarification.

1. My single experimental suggestion would be to more fully parse the cell autonomous and non-autonomous rolls of cerebellin-1 for DCNs. When Cerebellin-1 is specifically knocked out of DCNs, do they ever reach the floor plate, or do they not grow long enough to get there? Or what happens in an open book DiI assay?

Clarification Points:

1. At the beginning of the results, "differentially expressed genes" needs to be clarified by adding "over developmental time." As written now, it is quite vague.

2. Figure 1A: Motor neurons should also be highlighted in E12.5.

3. Figure 1B-D: Not really convinced by the LHx2 staining that Cbln1 is expressed in DCN (Figure 2A is more convincing). Adding zoomed insets may help?

4. Figure 3A-C: Old work from Tessier-Lavigne group showed no outgrowth of spinal explants, unless netrin was present. Was a modification to your system made? Commenting on this would be helpful.

F-H? worth using an additional inhibitor to nail this down a role for the lysosome or otherwise more clearly citing role for lysosome exocytosis in cerebellin-1 secretion from other papers for those not seeped in the field.?

5. Figure 6F: clarify in the text that this addition to wildtype, although would be interesting if rescued in knockout?

6. Figure 7C: Tag1 is marking axons of RGC---how were the regions defined, as it is unclear.

7. Figure 7D: Please include representative images.

8. Figure 7H: Why don't we see ipsilateral projection here? My understanding is that DiI labeling would also label the ipsilateral axons.

Other suggestions:

* Ibata et al shows that secreted Cbln1 binds Neurexin and diffuses along the axon. The authors show IF images of Cbln1 (Fig 3) and Neurexin (Fig 5) each localizing to commissural axons and the growth cone in cultured DCN neurons. May be useful to show if these proteins co-localize at the axon/growth cone in cultured neurons? Furthermore, does loss of Neurexin in cultured neurons decrease Cbln1 staining at the axon/growth cone?

* In the discussion, I would be interested in hearing the authors' thoughts on the other Cerebellin isoforms/any possible redundancy (or lack thereof) in axon guidance.

---

## [Editor Report · Decision Letter 2]

2 Sep 2022

Dear Dr Ji,

Thank you for your patience while we considered your revised manuscript entitled "Cbln1 regulates axon growth and guidance in multiple neural regions" for publication as a Research Article at PLOS Biology. This revised version of your manuscript has been evaluated by the PLOS Biology editors and the Academic Editor.

Based on the discussion, we are likely to accept this manuscript for publication, provided you satisfactorily address the data and other policy-related requests stated below.

We expect to receive your revised manuscript within two weeks. 

*Published Peer Review History*

*Press*

Sincerely,

Ines

--

Ines Alvarez-Garcia, PhD

Senior Editor,

ialvarez-garcia@plos.org,

PLOS Biology

ETHICS STATEMENT:

-- Please include the license number used.

Fig. 2D, E, H, I; Fig. 3B, C, E, G, H; Fig. Fig. 4B, C, F, G, I; Fig. 5D, F; fig. 6E, H; Fig. 7E, L; Fig. S2B, C, F-H, J, K; Fig. S3A, C; Fig. S4C, E, G, H and Fig. S5B-D, F

**Please also make the data you have deposited in GEO with accession number GSE169448 publicly available at this stage.

SPECIES INDICATED IN THE ABSTRACT? 

- Please note that per journal policy, the model system/species studied should be clearly stated in the abstract of your manuscript.

We require the original, uncropped and minimally adjusted images supporting all blot and gel results reported in an article's figures or Supporting Information files. We will require these files before a manuscript can be accepted so please prepare and upload them now. Please carefully read our guidelines for how to prepare and upload this data: https://journals.plos.org/plosbiology/s/figures#loc-blot-and-gel-reporting-requirements

---

## [Editor Report · Decision Letter 3]

27 Sep 2022

Dear Dr Ji,

Thank you for the submission of your revised Research Article entitled "Cbln1 regulates axon growth and guidance in multiple neural regions" for publication in PLOS Biology. On behalf of my colleagues and the Academic Editor, Franck Polleux, I am happy to say that we can in principle accept your manuscript for publication, provided you address any remaining formatting and reporting issues. These will be detailed in an email you should receive within 2-3 business days from our colleagues in the journal operations team; no action is required from you until then. Please note that we will not be able to formally accept your manuscript and schedule it for publication until you have completed any requested changes.

PRESS

Sincerely, 

Ines

--

Ines Alvarez-Garcia, PhD

Senior Editor

PLOS Biology
